# Allosteric role of the citrate synthase homology domain of ATP citrate lyase

Xuepeng Wei ®[1,2,6], Kollin Schultz[3], Hannah L. Pepper ®[4,5], Emily Megill[4,5], Austin Vogt[1,2], Nathaniel W. Snyder ®[4,5] & Ronen Marmorstein ®[1,2,3] ✉

ATP citrate lyase (ACLY) is the predominant nucleocytosolic source of acetyl-CoA and is aberrantly regulated in many diseases making it an attractive therapeutic target. Structural studies of ACLY reveal a central homotetrameric core citrate synthase homology (CSH) module flanked by acyl-CoA synthetase homology (ASH) domains, with ATP and citrate binding the ASH domain and CoA binding the ASH-CSH interface to produce acetyl-CoA and oxaloacetate products. The specific catalytic role of the CSH module and an essential D1026A residue contained within it has been a matter of debate. Here, we report biochemical and structural analysis of an ACLY-D1026A mutant demonstrating that this mutant traps a (3S)-citryl-CoA intermediate in the ASH domain in a configuration that is incompatible with the formation of acetyl-CoA, is able to convert acetyl-CoA and OAA to (3S)-citryl-CoA in the ASH domain, and can load CoA and unload acetyl-CoA in the CSH module. Together, this data support an allosteric role for the CSH module in ACLY catalysis.

ATP citrate lyase (ACLY) is a major source of nucleocytosolic acetyl-CoA, an essential building block for the biosynthesis of fatty acids, isoprenoids, and cholesterol, as well as a cofactor for acetylation of histones to regulate gene expression and other proteins and RNA to regulate diverse biological activities[1]. Aberrant activity of ACLY is found in many disorders[2], including nonalcoholic fatty liver disease[3,4], atherosclerotic cardiovascular disease[5,6], and cancers of the breast, prostate, cervix, lung, liver, and brain[7–10]. The importance of ACLY in health and disease has prompted a renewed interest in a detailed understanding of its molecular mechanism and the development of ACLY inhibitors for therapeutic application[2,8,10–13], including the identification of bempedoic acid, recently approved by the FDA to reduce LDL cholesterol for patients with atherosclerotic cardiovascular risk[11,14,15].

ACLY contains an N-terminal acyl-CoA synthetase homology (ASH) superdomain[16] and a C-terminal citrate synthase homology (CSH) domain[17,18]. Together, these domains use mitochondrial-derived citrate with Mg-ATP and Coenzyme-A (CoA) co-substrates to produce the acetyl-CoA and oxaloacetate (OAA) coproducts. Structural studies have been instrumental in providing important details underlying the molecular mechanism of ACLY activity. Pioneering X-ray crystallographic work from the Fraser laboratory revealed that the isolated ASH domain binds citrate and ATP to form a covalent citryl-phosphate adduct[19,20], but the ASH is unable to bind CoA on its own, thus implicating a role for CSH domain in CoA binding and subsequent ACLY catalysis. More recent biochemical and structural studies revealed that the CSH domain mediates homotetramerization of ACLY, whereby the CSH domains form a centrally located tetramerization module, with two each of the ASH domains on opposite sides of the CSH module[21–24]. The structures also show that different subunits of the tetramer participate in binding each of four CoA substrates, with the adenine base and ribose ring interacting with the CSH domain of one subunit and the pantothenic arm and β-mercapto group pointing into the active site of the ASH domain of another subunit. Paradoxically, the cryo-EM

[1]Department of Biochemistry & Biophysics, Perelman School of Medicine, University of Pennsylvania, Philadelphia, PA 19104, USA. [2]Abramson Family Cancer Research Institute, Perelman School of Medicine, University of Pennsylvania, Philadelphia, PA 19104, USA. [3]Graduate Group in Biochemistry & Molecular Biophysics, Perelman School of Medicine, University of Pennsylvania, Philadelphia, PA 19104, USA. [4]Department of Cardiovascular Sciences, Lewis Katz School of Medicine, Temple University, Philadelphia, PA 19140, USA. [5]Center for Metabolic Disease Research, Lewis Katz School of Medicine, Temple University, Philadelphia, PA 19140, USA. [6]Present address: GMU-GIBH Joint School of Life Sciences, The Guangdong-Hong Kong-Macau Joint Laboratory for Cell Fate Regulation and Diseases, Guangzhou Laboratory, Guangzhou Medical University, Guangzhou, China. ✉e-mail: marmor@upenn.edu

structure of ACLY with CoA revealed a second mutually exclusive CoA binding site where the ADP moiety is shifted about 8 Å towards the CSH module and the pantothenic arm is bent over to interact with the CSH module[24]. The significance of this alternative binding site was reinforced by a crystal structure of the isolated CSH module bound to CoA in an analogous fashion[22].

The presence of a "non-productive" CoA binding site against the CSH module has led to two alternative models for catalysis by ACLY (Fig. 1). In model 1, CoA, citrate, and ATP bind to the ACLY ASH domain to produce a (3S)-citryl-CoA intermediate. The ACLY enzyme then undergoes a structural rearrangement to translocate the intermediate to the CSH module, where cleavage of the intermediate occurs to form the acetyl-CoA and OAA products[22,23]. In model 2, CoA first binds to the ACLY CSH domain in a non-productive conformation, which induces a structural rearrangement of ACLY, resulting in the translocation of CoA to the ASH domain in a productive conformation, where CoA reacts with ASH-bound citrate and ATP to form OAA and acetyl-CoA products[24]. Once the acetyl-CoA product is formed in the ASH domain, another structural rearrangement of ACLY unloads acetyl-CoA

through binding to the CSH module before acetyl-CoA leaves ACLY[24]. Consistent with model 2, the cryo-EM structure of ACLY in the presence of acetyl-CoA and OAA products reveal that they are bound in the ASH domain[24]. The difference between the two models of ACLY catalysis centers around the role of the CSH domain in ACLY catalysis. In model 1, residues within the CSH domain are proposed to play a direct role in cleavage of (3S)-citryl-CoA to acetyl-CoA and OAA products with residue Asp1026, shown to be essential for ACLY catalysis, acting as a general base for the reaction[22,23]. In model 2, the CSH module plays an allosteric role in binding the CoA substrate prior to its translocation to the ASH domain and binding the acetyl-CoA product prior to release from the enzyme[24]. Consistent with the importance of Asp1026 in ACLY function, mutation of this residue to alanine renders the enzyme inactive[24,25].

To help resolve the alternative models on the role of the CSH module on ACLY catalysis, we report here a biochemical and structural analysis of the ACLY-D1026A mutant. Our data is most consistent with an allosteric role (model 2) of the CSH module in ACLY catalysis.

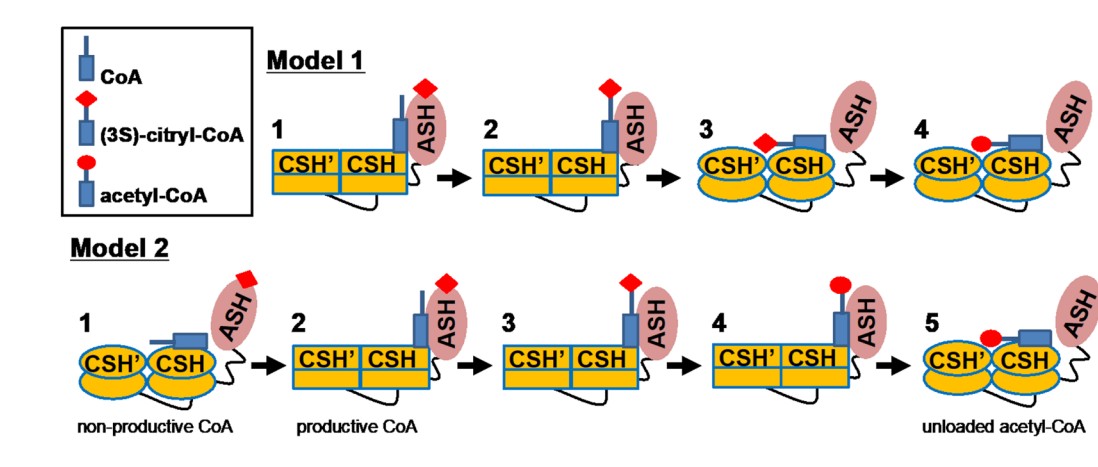

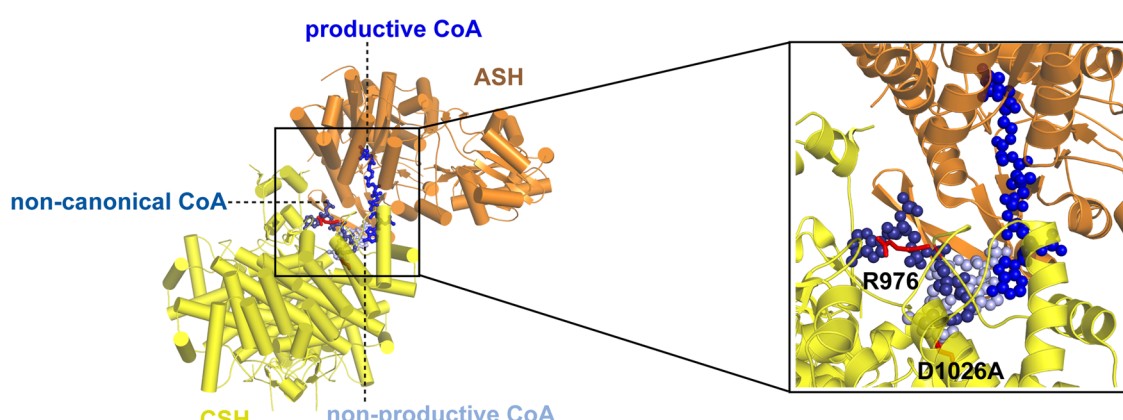

**Fig. 1 | Models for ACLY catalysis and CoA binding modes. a** Model 1—ACLY-WT in complex 1 binds CoA through the ASH domain to form (3 S)-citryl-CoA in the ASH domain (complex 2). (3 S)-citryl-CoA then translocates to the CSH domain where catalysis to acetyl-CoA occurs (complex 4). Model 2—ACLY-WT in complex 1 loads CoA onto the CSH domain in a non-productive conformation with the ASH domain in an open conformation. An allosteric change in the CSH domain, coupled with a conformational change of the ASH domain to a closed conformation (complex 2), mediates the translocation of CoA to the ASH domain. Complex 3 initiates catalysis by forming (3S)-citryl-CoA in the ASH domain, followed by complex 4, that completes catalysis to acetyl-CoA in the ASH domain. An

allosteric change in the CSH domain, coupled with a conformational change of the ASH domain to an open conformation (complex 5), mediates the translocation of acetyl-CoA to the CSH domain prior to product release. Different bound states of CoA and acetyl-CoA are annotated below the corresponding schematic diagrams. Residue D1026 is located in the CSH domain and interacts with non-productive CoA. Only one of the four ASH domains of the tetramer is shown, and the binding of ATP and OAA to the ASH domain is not shown for clarity. The red diamond and circle represent citrate and acetate, respectively. **b** Comparison of productive and non-productive CoA binding modes highlighting key residues highlighted in this study.

## Results

### ACLY-D1026A with substrates traps (3S)-citryl-CoA in the ASH domain

To determine how the ACLY-D1026A mutant engages substrates, we preincubated ACLY-D1026A with substrates (citrate, CoA, and ATP) in an attempt to capture a reacted state of the mutant enzyme for cryo-EM structure determination. The cryo-EM structure (ACLY-D1026A−substrates) was determined at an overall resolution of 2.2 Å with imposing D2 symmetry (Fig. 2a, Table 1, and Supplementary Fig. 1). The symmetric structure showed the expected tetrameric architecture of ACLY indicating that the mutant protein did not contain gross structural alterations when compared to the ACLY-WT structure bound to CoA (PDB 6UUZ, RMSD = 1.25)[24] (Fig. 2a). The CSH module had a local resolution of between 1.9–2.4 Å, while the ASH domain was more flexible, with local resolution between 2.5–3.4 Å (Fig. 2a, b), also consistent with the trend seen with the previously reported cryo-EM ACLY structures with and without bound ligands[23,24].

Similar to the reported ACLY-WT−CoA structure (6UUZ)[24], the ACLY-D1026A−substrates structure revealed well-ordered cryo-EM density for the phosphorylated ADP group of CoA bound at the interface between ASH and CSH domain with additional density pointing into the ASH domain, which could best be modeled as a (3S)-citryl-CoA reaction intermediate (Fig. 2c and Supplementary Fig. 2A). Unexpectedly, we observed unaccounted for cryo-EM density against the CSH domain that could be modeled as an additional CoA molecule bound in a previously unobserved "non-canonical" conformation, with the ADP group sitting near the center of the CSH module and the thioester of CoA pointing towards the end of the CSH module and bound in a 'hole' that is created by the removal of Asp1026 (Fig. 2c). We

hypothesize that the ACLY-D1026A mutation disrupts the water-mediated interaction with the sulfur atom of CoA, and residues H900 and R1065 that stabilize binding of non-productive CoA[24], thus making the site available for mutually exclusive binding of non-canonical CoA. Notably, Arg 976, which plays an important role in binding non-productive, CoA changes position to avoid a steric clash with the binding of non-canonical CoA (Supplementary Fig. 3). The non-canonical CoA molecule bound to the CSH domain of the ACLY-D1026A−substrates structure sits across two subunits of the tetrameric CSH module (composed of four CSH domain subunits) such that it forms hydrophobic interactions with one CSH domain subunit and hydrogen bonds with the other CSH domain subunit, potentially locking the CSH module in a "closed" conformation that would not accommodate a translocated product from the ASH domain (Fig. 2d, E). In addition, a translocated product from the ASH domain would have to displace the non-canonical CoA molecule bound to the CSH domain. It therefore appears that the non-canonical CoA serves to trap (3 S)-citryl-CoA within the ASH domain.

### ACLY-D1026A with (3S)-citryl-CoA in the ASH domain is incompatible with the formation of products

In the ACLY reaction mechanism, His760 transiently accepts the phosphate group from ATP prior to transfer to citrate to form a phosphocitrate intermediate[26]. Prior crystallographic studies and our recent cryo-EM studies reveal that the loop containing His760 is only ordered when phospho-histidine or a mimic is present[24,27]. Interestingly, the loop containing His760 is disordered in the ACLY-D1026A−substrates structures, even though phosphate in the form of ATP was added in the ACLY-D1026A−substrates structure. These observations

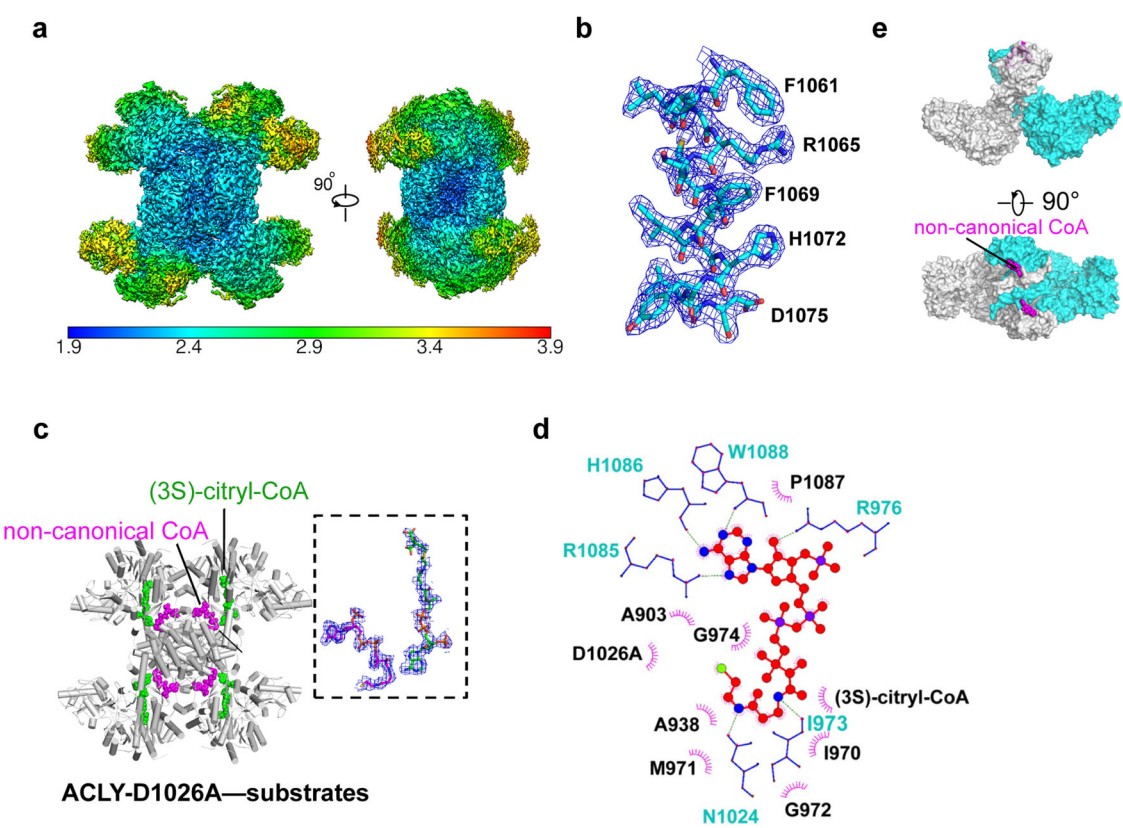

**Fig. 2 | Structure of ACLY-D1026A with substrates. a** Cryo-EM map and local resolution estimation of ACLY-D1026A−substrates. **b** Representative side chain cryo-EM density of residues in the CSH domain. **c** Cartoon presentation of ACLY-D1026A−substrates highlighting bound non-canonical CoA (purple) and (3S)-citryl-CoA (green) molecules. **d** Ligplot highlighting interactions between the non-canonical CoA, two adjacent CSH domains, and (3 S)-citryl-CoA. **e** Surface representation of ACLY-D1026A−substrates structure highlighting how the non-canonical CoA molecule (magenta) interacts with the adjacent CSH domain; for clarity, only two subunits (cyan and gray) are shown.

**Table 1 | Cryo-EM data collection, refinement, and validation statistics**

| | D1026A—substrates (EMD-24479) (PDB- 7RIG) | D1026A—substrates, local refinement of ASH domain (EMD-29739) (PDB- 8G5C) | D1026A—substrates- asym-int (EMD-24511) (PDB- 7RKZ) | D1026A—substrates- asym (EMD-24577) (PDB- 7RMP) | D1026A—products (EMD-29669) (PDB- 8G1F) | D1026A—products, local refinement of ASH domain (EMD-29740) (PDB- 8G5D) | D1026A—products-asym (EMD-29668) (PDB- 8G1E) |
|---|---|---|---|---|---|---|---|
| **Data collection and processing** | | | | | | | |
| Magnification | 105,000 | 105,000 | 105,000 | 105,000 | 105,000 | 105,000 | 105,000 |
| Voltage (kV) | 300 | 300 | 300 | 300 | 300 | 300 | 300 |
| Electron exposure (e–/Å$^2$) | 40 | 40 | 40 | 40 | 40 | 40 | 40 |
| Defocus range (µm) | 1.0–2.0 | 1.0–2.0 | 1.0–2.0 | 1.0–2.0 | 1.0–2.0 | 1.0–2.0 | 1.0–2.0 |
| Pixel size (Å) | 0.83 | 0.83 | 0.83 | 0.83 | 0.83 | 0.83 | 0.83 |
| Movies (no.) | | | 5706 | | | 6057 | |
| Initial particle images (no.) | | | 2,994,064 | | | 2,126,238 | |
| Final particle images (no.) | 378,979 | 1,122,938 | 237,362 | 100,062 | 289,796 | 613,268 | 183,036 |
| Symmetry imposed | D2 | C1 | C1 | C1 | D2 | C1 | C1 |
| Map resolution (Å) | 2.2 | 2.2 | 2.6 | 2.7 | 2.4 | 2.5 | 2.8 |
| FSC threshold | 0.143 | 0.143 | 0.143 | 0.143 | 0.143 | 0.143 | 0.143 |
| **Refinement** | | | | | | | |
| Initial model used (PDB code) | 6uuw | 6uuw | 6uuw | 6uuw | 6uuw | 6uuw | 6uuw |
| Model resolution (Å) | 2.3 | 2.4 | 2.7 | 3.0 | 2.6 | 2.6 | 3.0 |
| FSC threshold | 0.5 | 0.5 | 0.5 | 0.5 | 0.5 | 0.5 | 0.5 |
| Map sharpening $B$ factor (Å$^2$) | –67.2 | –63.2 | –75.6 | –74.0 | –87.8 | –83.2 | –93.9 |
| Model composition | | | | | | | |
| Non-hydrogen atoms | 32,803 | 14,874 | 32,610 | 32,553 | 32,911 | 14,852 | 32,649 |
| Protein residues | 4128 | 1869 | 4126 | 4128 | 4148 | 1874 | 4148 |
| Ligands | 22 | 12 | 16 | 19 | 22 | 12 | 18 |
| R.m.s. deviations | | | | | | | |
| Bond lengths (Å) | 0.006 | 0.003 | 0.003 | 0007 | 0.008 | 0.005 | 0.006 |
| Bond angles (°) | 0.691 | 0.531 | 0.691 | 0.746 | 0.767 | 0.739 | 0.666 |
| Validation | | | | | | | |
| MolProbity score | 1.61 | 1.63 | 1.89 | 1.84 | 1.65 | 1.81 | 1.77 |
| Clashscore | 10.27 | 9.67 | 14.09 | 13.18 | 8.12 | 11.24 | 9.73 |
| Poor rotamers (%) | 0.88 | 1.04 | 0.26 | 0.23 | 0.50 | 0.39 | 0.18 |
| Ramachandran plot | | | | | | | |
| Favored (%) | 97.66 | 98.17 | 96.32 | 96.64 | 97.68 | 96.83 | 96.2 |
| Allowed (%) | 2.34 | 1.83 | 3.68 | 3.36 | 2.32 | 3.17 | 3.80 |
| Disallowed (%) | 0 | 0 | 0 | 0 | 0 | 0 | 0 |
| EMRinger score | 3.93 | 4.83 | 3.09 | 3.05 | 3.43 | 3.65 | 3.01 |

are consistent with the ACLY-WT-like hydrolysis of ATP in the ACLY-D1026A mutant, but the non-typical conformation of the ACLY active site that might contribute to the trapping of (3S)-citryl-CoA in the active site.

Notably, a superposition of ASH domains from the ACLY-D1026A−substrates and ACLY−OAA−acetyl−CoA product structure (6UV5) reveals that the (3S)-citryl-CoA bound to the ASH domain of the ACLY-D1026A−substrates structure is incompatible with cleavage to the acetyl-CoA and OAA products. Specifically, Phe347 in the ASH appears to take on two distinct conformations when the ASH domain is bound to (3S)-citryl-CoA or acetyl-CoA + OAA (Fig. 3a, b). When bound to (3S)-citryl-CoA, Phe347 is oriented towards the pantetheine arm to make van der Waals interactions with the ligand and block the binding site of OAA (Fig. 3b–d). When bound to acetyl-CoA and OAA, Phe347 is flipped away from acetyl-CoA to accommodate the bound OAA molecule

(Fig. 3b). Taken together, it appears that the formation of (3S)-citryl-CoA in the ASH with the accompanying structural rearrangement of Phe347 prevents the ordered production and release of acetyl-CoA and OAA products in the ACLY-D1026A mutant.

**Asymmetric ACLY subunit shows allosteric regulation of the ASH domain**

An ACLY-D1026A−substrates complex refined without symmetry (ACLY-D1026A−substrates-asym) reveals three symmetric and one asymmetric conformation (Table 1 and Supplementary Fig. 2B). The symmetric ASH domains reveal an ASH active site and corresponding CSH domain containing bound (3S)-citryl-CoA and non-canonical CoA, respectively, as in the symmetric structure (Fig. 2c). Multiple rounds of 3D hetero refinement classification on the asymmetric particles (ACLY-D1026A−substrates-asym-int and ACLY-D1026A-substartes-asym in

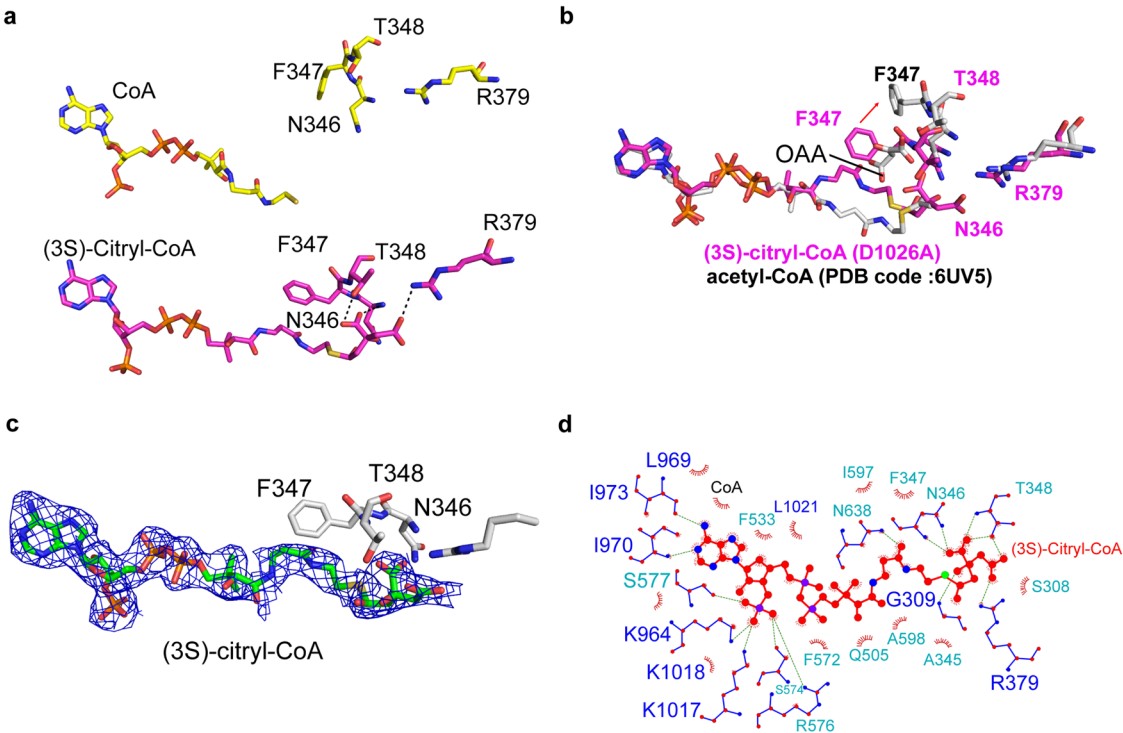

**Fig. 3 | (3S)-Citryl-CoA binding sites in the ASH domain of ACLY-D1026A with substrates. a** Comparison of CoA and (3S)-Citryl-CoA bound to the ASH domain of ACLY-WT bound to CoA (PDB 6UUZ) and ACLY-D1026A−substrates structures, respectively. Residues that change positions to accommodate the different ligands are highlighted. **b** Overlay of ACLY-D1026A−substrates and ACLY-WT bound to acetyl-CoA + OAA (PDB 6UV5) highlighting the conformational change of F347. **c** Cryo-EM density of (3 S)-citryl-CoA in the ASH domain is displayed at a contour level of 2.5 σ. **d** A ligplot highlighting the interactions between (3 S)-citryl-CoA and surrounding residues within the ASH domain.

Table 1) revealed a large amount of conformational flexibility within this domain. The asymmetric ASH domain did not contain traceable cryo-EM density for a bound ligand, but instead, the CSH domain did contain cryo-EM density that could be tentatively modeled as CoA + citrate, although (3 S)-citryl-CoA could also be modeled (Table 1 and Supplementary Fig. 2C). Notably, the modeled CoA bound to the CSH domain is in the 'loaded' position of non-productive CoA in the structure of ACLY-WT with CoA and citrate (PDB 6UUZ)[24]. A superposition of the ASH-bound (3 S)-citryl-CoA/CSH-bound non-productive CoA pair with the CSH-bound non-canonical CoA reveals the mutually exclusive nature of these two binding states (Supplementary Fig. 2D). The relative positions of these ligands is consistent with CoA bound to the CSH in a pre-translocated loaded position, which moves to the ASH for the formation of the (3 S)-citryl-CoA intermediate, and that bound non-canonical CoA serving to trap (3 S)-citryl-CoA within the ASH domain. If the ligand bound to the CSH domain is (3 S)-citryl-CoA, this could represent an "unloaded" (3 S)-citryl-CoA, which could not form acetyl-CoA and OAA products in the ASH. In either case, this asymmetric structure is consistent with the allosteric regulation of the ASH domain for ACLY catalysis.

### ACLY-D1026A with products is able to form (3S)-citryl-CoA in the ASH domain

Given the findings above that the ACLY-D1026A mutant is not config-ured to form acetyl-CoA and OAA products in the ASH domain, we reasoned that if we saturated the mutant with acetyl-CoA and OAA products, that we would either find acetyl-CoA bound to the CSH domain (in its unloaded position) or converted to citryl-CoA within the ASH domain, since the retro-aldol cleavage reaction is reversible. We proceeded to incubate the ACLY-D1026A mutant with saturating levels of acetyl-CoA, OAA, and ATP (to more closely mimic an active con-formation of the H760-containing loop), incubated at room

temperature for 15 min, and determined the structure by cryo-EM. The ACLY-D1026A−products structure was determined to have an overall resolution of 2.4 Å imposing D2 symmetry and to 2.8 Å without sym-metry imposed. After building and refining the symmetric structure, we observed unambiguous density in the ASH domain, which could best be modeled as (3S)-citryl-CoA as well as a non-canonical acetyl-CoA bound to the CSH domain (similar to the non-canonical CoA bound to the CSH domain of the ACLY-D1026A−substrates structure) (Fig. 4a−c, Table 1, and Supplementary Fig. 4). A comparison of the protein environment around the (3S)-citryl-CoA, reveals a striking similarity to the environ-ment around (3S)-citryl-CoA in the ACLY-D1026A−substrates complex (Fig. 4d). In particular, Phe347 adopts the same conformation that blocks OAA binding. Interestingly, the loop containing His760 is well resolved in the ACLY-D1026A−products structure, indicating that the ACLY-D1026A-product structure adopts a slightly different conforma-tion than the ACLY-D1026A−substrates structure. Together, this finding reveals that the ASH domain contains the catalytic residues for the conversion of (3S)-citryl-CoA to acetyl-CoA and OAA.

Like the asymmetric ACLY-D1026A−substrates-asym structure, the ACLY-D1026A−products-asym structure reveals three symmetric and one asymmetric and more flexible ASH domain with more varied conformation (Table 1 and Supplementary Figs. 4, 5A, B). The asym-metric ASH domain is missing traceable cryo-EM ligand density in the ASH domain, but instead contains cryo-EM density along the corre-sponding CSH domain that can be modeled as acetyl-CoA in an "unloading" configuration plus OAA (Supplementary Fig. 5C). A superposition of ASH-bound (3 S)-citryl-CoA/CSH-bound unloading acetyl-CoA pair with the non-canonical acetyl-CoA reveals the mutually exclusive nature of these two binding states (Supplementary Fig. 5D). The relative positions of these ligands are consistent with acetyl-CoA product formation in the ASH domain followed by unloading of acetyl-CoA through the CSH domain.

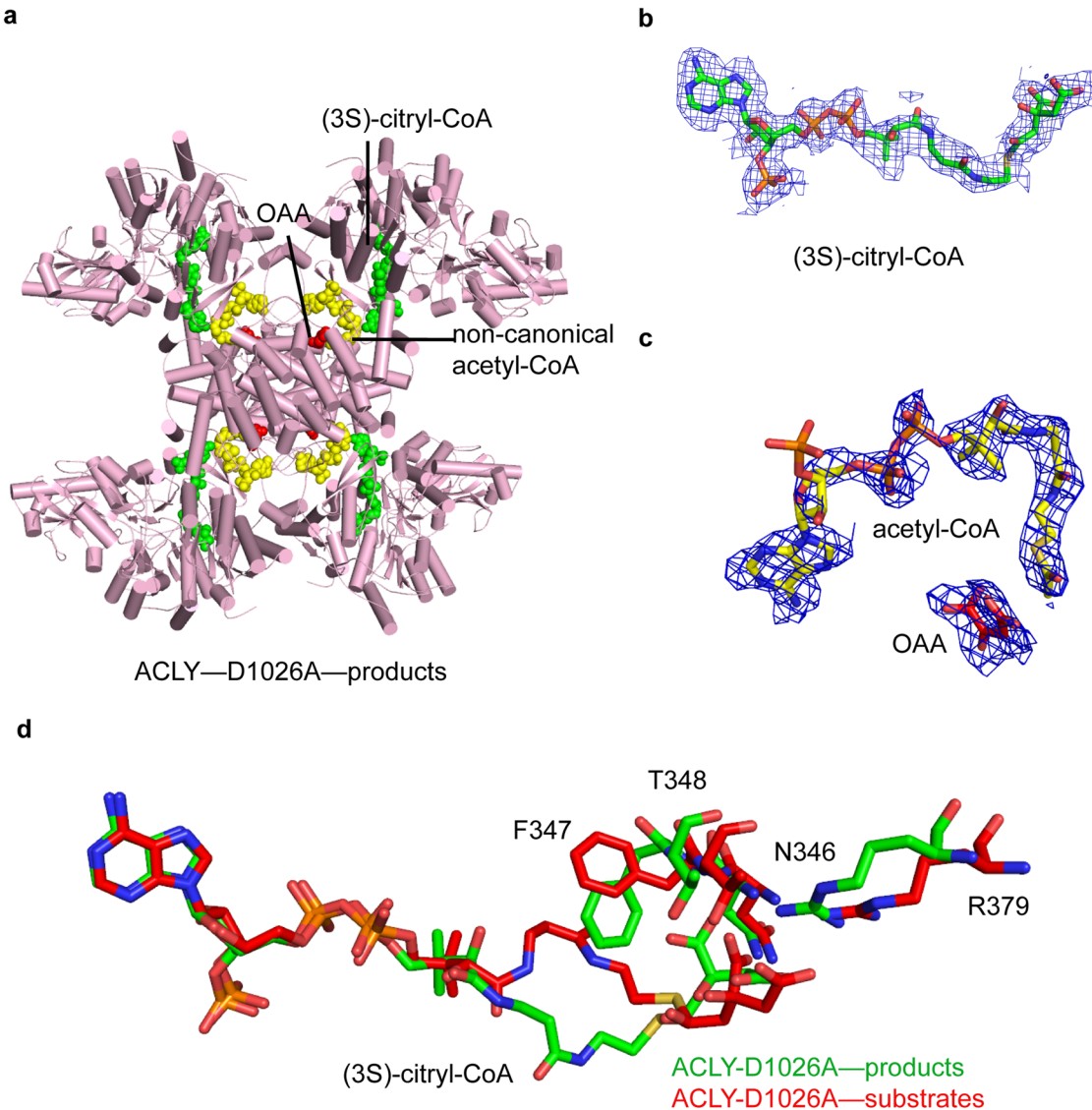

**Fig. 4 | Structure ACLY-D1026A with products. a** Cartoon presentation of ACLY-D1026A−products structure highlighting bound (3S)-citryl-CoA (green), non-canonical acetyl-CoA (yellow), and OAA (red) molecules. **b** Cryo-EM density of (3S)-citryl-CoA bound at the ASH domain. **c** Cryo-EM density of non-canonical acetyl-CoA and OAA bound at the CSH domain. **d** Overlay of (3S)-citryl-CoA from ACLY-D1026A−products and ACLY-D1026A−substrates structures.

To confirm the formation of the (3S)-citryl-CoA intermediate by the ACLY-D1026A mutant in solution, we incubated the mutant protein with acetyl-CoA, OAA, ATP (to more closely mimic an active conformation of the H760-containing loop) and 5 mM MgCl₂ at room temperature for 15 min, quenched the sample with 10% trichloroacetic acid in water, and analyzed the acyl-CoAs in the reaction by liquid chromatography-quadrupole/Orbitrap high-resolution mass spectrometry (LC-HRMS and LC-MS/MS)[28]. We identified citryl-CoA within the extracts from the enzymatic reaction with the ACLY-D1026 A mutant, while analogous reactions without OAA and ATP or without ATP did not result in detectable levels of citryl-CoA (Fig. 5 and Supplementary Fig. 6A, B). In an additional control reaction, ACLY-WT incubated with acetyl-CoA, OAA, and ATP did not show detectable levels of citryl-CoA, presumably because citryl-CoA forms only transiently in ACLY-WT (Supplementary Fig. 6C). Interestingly, ACLY-D1026A incubated with acetyl-CoA, OAA, and ATPγS also produced citryl-CoA, while substitution of ATP/ ATPγS with ADP or ADP + phosphate did not show detectable levels of citryl-CoA (Supplementary Fig. 6C), suggesting that an ATP-induced conformational change of the H760-containing loop is required for citryl-CoA formation from OAA and acetyl-CoA

products. These solution studies support our structural observation that the ACLY-D1026A mutant is able to convert acetyl-CoA and OAA substrates to citryl-CoA products within the ASH domain.

To demonstrate that citryl-CoA formation by the ACLY-D1026A mutant is a kinetically competent intermediate, we incubated ACLY-D1026A with acetyl-CoA and OAA products and ATP and 5 mM MgCl₂ at room temperature and monitored citryl-CoA formation over time by quenching the reaction and semi-quantitation of citryl-CoA formation by LC-HRMS and LC-MS/MS using $^{13}C_3$ $^{15}N_1$-HMG-CoA as a surrogate internal standard to derive a relative abundance of citryl-CoA. (Supplementary Fig. 7A). This analysis revealed a time-dependent increase of citryl-CoA formation with ACLY-D1026A but not with ACLY-WT. Analogous studies of ACLY-D1026A and ACLY-WT incubated with CoA, citrate and ATP substrates also revealed time-dependent formation of citryl-CoA for ACLY-D1026A, but not for ACLY-WT (Supplementary Fig. 7B). ACLY-WT, but not ACLY-D1026A, showed time-dependent formation of acetyl-CoA. These studies demonstrate that the ACLY-D1026A mutant produces a kinetically competent citryl-CoA intermediate when incubated with either CoA and citrate substrates or acetyl-CoA and OAA products.

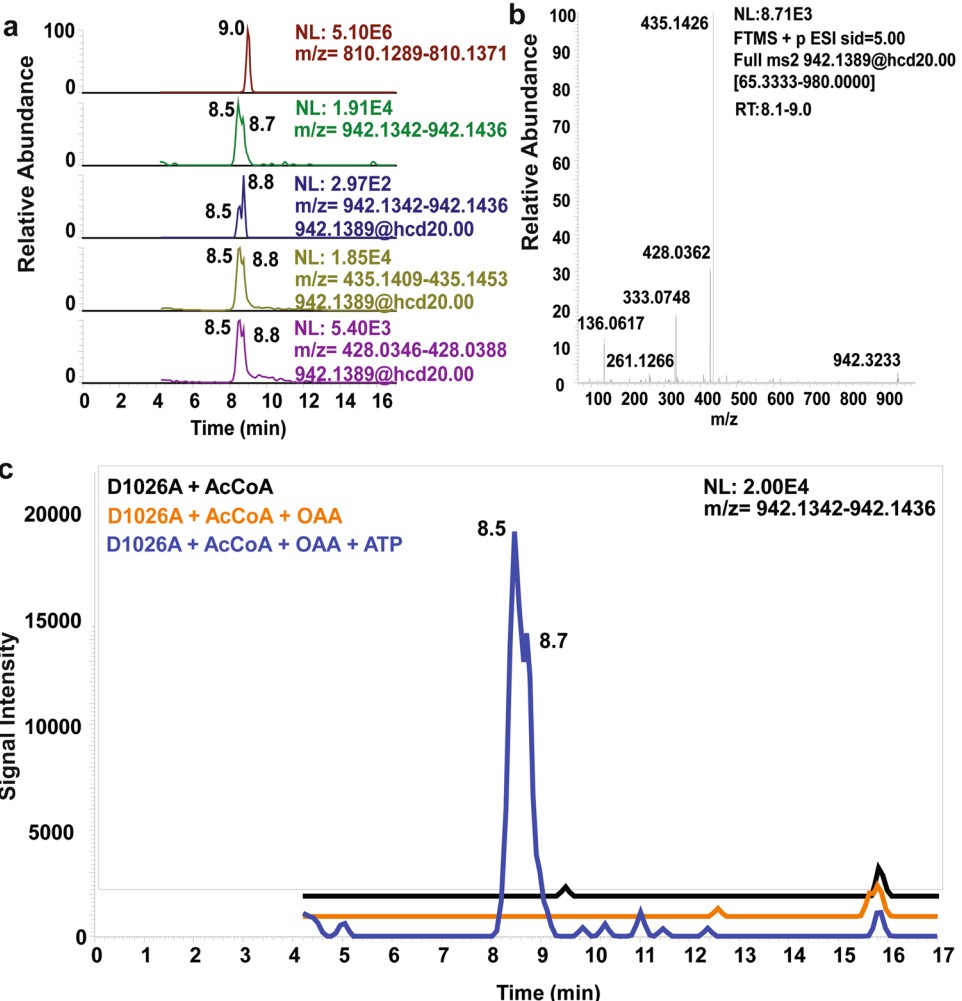

**Fig. 5 | Identification of citryl-CoA by LC-HRMS and LC-MS/MS produced by ACLY-D1026A. a** Unreacted acetyl-CoA (top, in red retention time 9.0, [M + H]⁺ ion at *m/z* 810.1330) and citryl-CoA (retention time 8.5-8.8, [M + H]⁺ ion at m/z 942.1389). Co-eluting MS/MS peaks corresponding to predominant acyl-CoA peaks at m/z 428.0362 and the [M-507 + H]⁺ neutral loss diagnostic for acyl-CoAs at 435.1426 support the identification of citryl-CoA. **b** MS/MS spectra for citryl-CoA.

**c** LC-HRMS chromatograms of the citryl-CoA product from ACLY-D1026A + AcCoA (black), ACLY-D1026A + AcCoA + OAA (orange), and ACLY-D1026A + AcCoA + OAA + ATP (blue). An intense peak corresponding to citryl-CoA was identified in the only extract from ACLY-D1026A + AcCoA + OAA + ATP with the baseline offset for clarity.

To further validate that altering the non-productive CoA binding site, independent of the D1026A mutation, could also lead to the formation of (3 S)-citryl-CoA from reaction products, we prepared an additional ACLY mutant that is predicted to destabilize non-productive CoA binding but not affect productive CoA binding and incubated this mutant with products and carried out similar LC-MS studies as described above to probe for production of (3 S)-citryl-CoA intermediate. The mutant that we prepared was ACLY-R976E within the CSH domain, as R976 was observed to interact with non-productive CoA (but not productive CoA) in the ACLY-WT bound to CoA refined without symmetry in the closed conformation, and an ACLY-R976E was shown to have background levels of activity[24]. Like ACLY-D1026A, we found that ACLY-R976E was able to generate citryl-CoA (even in the presence of residue D1026) (Supplementary Fig. 8). Together, our data demonstrate that two different mutants that are predicted to disrupt non-productive CoA binding produce detectable (3 S)-citryl-CoA intermediate from products. With our structural observation of (3 S)-citryl-CoA bound to the ASH domain, this data further supports the conclusion that conversion of (3 S)-citryl-CoA to acetyl-CoA and OAA occurs in the ASH domain.

## Discussion

The homotetrameric ACLY enzyme is composed of a central homo-tetrameric CSH module with two ASH modules on opposite sides of the CSH module. It is well established that citrate and ATP bind to the ASH domain with CoA binding at the ASH-CSH domain interface to generate acetyl-CoA and OAA products. However, there have been contradictory reports on the fate of the (3 S)-citryl-CoA reaction intermediate and the role of the CSH domain in processing this intermediate (Fig. 1). In model 1, the (3 S)-citryl-CoA intermediate is translocated from the ASH domain to the CSH domain where the CSH domain participates in cleavage of the intermediate to the acetyl-CoA and OAA products. In model 2, the CSH domain participates in loading CoA from a non-productive to productive conformation into the ASH domain, followed by (3 S)-citryl-CoA formation and cleavage to the acetyl-CoA and OAA products in the ASH domain, and finally the release of acetyl-CoA through the CSH domain. In this case, the CSH domain plays an allosteric role in ACLY catalysis.

This study was carried out to help resolve these alternative models underlying the role of the CSH module of ACLY in catalysis, by focusing on an inactive D1026A mutant within the ACLY CSH domain. In model 1, D1026 is proposed to act as a general base for cleavage of

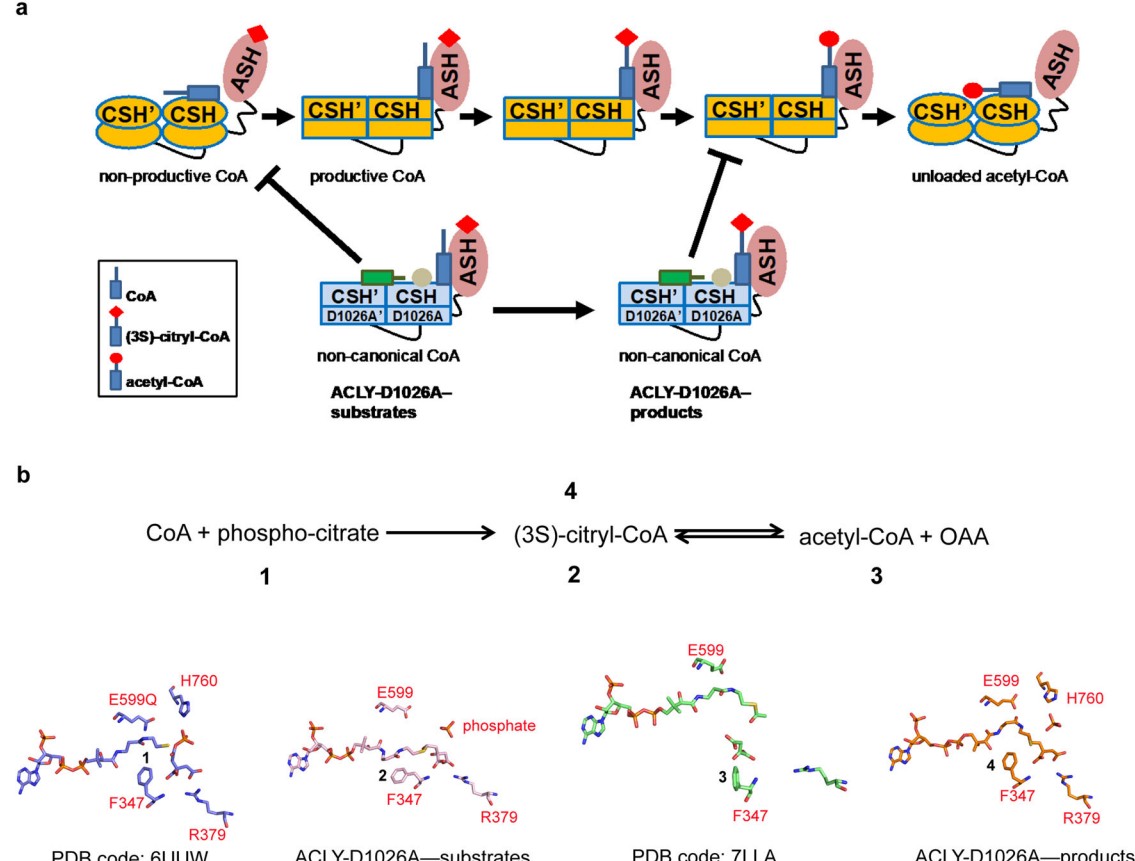

**Fig. 6 | Allosteric regulation of the CSH domain in ACLY catalysis and inhibition of product formation in ACLY-D1026A. a** Model 2 for ACLY catalysis, as illustrated in Fig. 1, with the mechanism of inhibition by the ACLY-D1026A mutant based on the structures of ACLY-D1026–substrates and ACLY-D1026A–products. Different bound states of CoA and acetyl-CoA are annotated below the corresponding schematic diagrams, and the position of the D1026A mutation is highlighted with a brown dot. The red diamond and circle represent citrate and acetate, respectively. **b** The reversible nature of the transformation between (3S)-citryl-CoA and acetyl-CoA and structurally captured states are indicated and shown, highlighting residues that undergo key conformational changes.

the (3 S)-citryl-CoA intermediate to products in the CSH domain, while in model 2, this mutant is proposed to disrupt the allosteric regulation required for ACLY activity, which involves CoA loading through the CSH domain, acetyl-CoA and OAA production in the ASH domain and acetyl-CoA unloading through the CSH domain. The cryo-EM structure ACLY-D1026A in complex with substrates (ACLY-D1026A−substrates) reported here demonstrates that the mutant alters the ASH active site such that the (3 S)-citryl-CoA intermediate cannot be cleaved into acetyl-CoA and OAA products, most likely due to the improper loading of non-productive CoA to the CSH domain (Fig. 6a). In particular, an important catalytic residue for acetyl-CoA formation, E599, is not in position for catalysis and F347 overlaps where OAA product is formed (Figs. 3b, 6b). Indeed, it appears that the conformational flexibility of F347, plays a key role in ACLY catalysis. The cryo-EM structure ACLY-D1026A in complex with acetyl-CoA and OAA products (ACLY-D1026A−products) also reveals formation of (3 S)-citryl-CoA in the ASH domain, demonstrating that the ASH domain contains the catalytic residues required to catalyze the reverse reaction, and thus capable of converting substrates to acetyl-CoA and OAA products. Formation of (3 S)-citryl-CoA from acetyl-CoA and OAA in the ACLY-D1026A mutant was also confirmed in solution through metabolomic analysis of the ACLY-D1026A mutant incubated with products and ATP (Fig. 5). We also detected the formation of (3 S)-citryl-CoA from acetyl-CoA in an ACLY-R976E mutant, which retains residue D1026 (Supplementary Fig. 8), further suggesting that transformation of (3 S)-citryl-CoA to acetyl-CoA can occur independently of residue D1026. Interestingly, our metabolic studies of ACLY-D1026A with acetyl-CoA and OAA

products does not form (3 S)-citryl-CoA in the absence of ATP (or ATPγS), consistent with the importance of a phosphate group and ordering of the H760-containing loop for catalysis. Taken together, these data demonstrate that the CSH domain does not play an essential catalytic role and is most consistent with model 2, whereby the CSH domain plays an allosteric role in ACLY catalysis.

The ACLY-WT product structure with acetyl-CoA and OAA (PDB 6UI9)[24], shows a second OAA molecule bound in a 'non-productive' OAA conformation to the CSH domain, which is proposed to play a regulatory role, potentially functioning as an inhibitor to inactivate ACLY at high OAA product concentrations[24]. This non-productive OAA was proposed to function by sequestering the pantothenic arm of CoA or acetyl-CoA to prevent proper loading and unloading, respectively[24]. Consistent with the functional importance of this non-productive OAA site, the ACLY-D1026A−product structure also contains cryo-EM density at the same site on the CSH domain, which we have also modeled as an OAA molecule (Fig. 4). In this structure, the acetyl-CoA and OAA interact through hydrogen bound between the acetyl-CoA acetyl group carbonyl and an OH of OAA, and van der Waals interactions between the acetyl methyl group and aliphatic region of OAA. This could explain how the non-productive OAA molecule could help stabilize acetyl-CoA binding to the CSH domain to prevent proper loading of CoA substrate to the CSH domain, thus inhibiting ACLY catalysis.

While this study does not exclude the possibility that the CSH domain of ACLY could directly participate in catalysis in some cellular context or in ACLY enzymes from other species, this study demonstrates that human ACLY can mediate acetyl-CoA production with

allosteric instead of direct catalytic participation of the CSH domain. Given the interest in developing ACLY inhibitors to treat many disorders, including hypercholesterolemia and cardiovascular disease, targeting the CSH module of ACLY may provide another viable site for drugging ACLY.

## Methods

### Expression and purification of proteins

A gene encoding human ACLY-WT with a C-terminal 6xHis tag, codon optimized for expression in bacteria (Genscript)[21] was used to prepare the ACLY-D1026A and ACLY-R576A mutants, via quick change mutagenesis[29]. Proteins were overexpressed in BL21(DE3) cells and purified to homogeneity using a combination of Cobalt Chelating Resin and gel filtration chromatography essentially as described for ACLY-WT[24]. The purified protein in a buffer containing 25 mM Hepes, pH7.5, 150 mM NaCl, 5 mM $MgCl_2$, and 10 mM β-mercaptoethanol was concentrated to 6–10 mg/mL, aliquoted into 20 uL portions and flash frozen in liquid nitrogen for storage in −80 freezer until use.

### LC-quadrupole/Orbitrap high-resolution mass spectrometry

ACLY samples for LC/MS analysis were assembled with 10 μM protein supplemented with 5 mM substrates (citrate, CoA, and ATP), or products (oxaloacetate, Acetyl-CoA, and ATP in place of ADP), as indicated, in 25 mM Hepes, pH7.5, 150 mM NaCl, 5 mM $MgCl_2$, and 10 mM β-mercaptoethanol. Reactions were allowed to proceed for 15 min for single time point experiments or between 0 and 60 min for time course experiments, before quenching reactions with 10% trichloroacetic acid in water and flash freezing to halt the reaction until LC/MS analysis. Acyl-CoA extractions and analysis to resolve acyl-CoA thioesters were performed as described previously in ref. 28. Briefly, extracts of enzymatic reactions were resuspended in 1000 μl of ice-cold 10% trichloroacetic acid and pulse-sonicated using a sonic dismembrator. The samples were centrifuged at $17,000 \times g$ for 10 min and the supernatants were purified by solid-phase extraction. Waters (Milford, MA) Oasis HLB 1 mL (30 mg) solid-phase extraction columns were conditioned with 1 mL methanol, followed by 1 mL of $H_2O$. The supernatants were applied to the column and washed with 1 mL of $H_2O$. The analytes were eluted in methanol containing 25 mM ammonium acetate. The eluates were evaporated to dryness under $N_2$ gas and resuspended in 50 μl of 5% 5-sulfosalicylic acid. About 10 μl injections were injected onto a Waters HSS T3 2.1 × 150 mm 3.5 μm Column on an Ultimate 3000 UHPLC coupled to a QExactive Plus (Thermo Fisher Scientific, San Jose, CA) operating in positive ion mode. Mass spectrometer settings were a resolution setting of 280,000, AGC target of 5e6, isolation window of 20 m/z with an offset of 6 m/z, and an insource-CID of 5 eV with data acquisition using XCalibur 4.3 and analysis with TraceFinder 5.1. The LC gradient was buffer A (5 mM ammonium acetate in water) in buffer B (5 mM ammonium acetate in 95:5 acetonitrile: water (v/v)) starting at 0.2 mL/min 100% A until 3 min, then 20% B at 5 min, then 100% B at 12 min, followed by washing with buffer C (acetonitrile: water: formic acid (80:20:0.1, v/v/v)) for 3.5 min at 0.275 mL/min then re-equilibration for 5 min at starting conditions. For semi-quantitation of citryl-CoA and acetyl-CoA during time course experiments, internal standards prepared biosynthetically from $^{13}C_3,^{15}N_1$-pantothenate were generated as previously described in ref. 30, spiked into each sample before extraction, and then $^{13}C_3,^{15}N_1$-HMG-CoA and $^{13}C_3,^{15}N_1$-acetyl-CoA, respectively, were used as internal standards to express relative abundance. Data were tabulated using Microsoft Excel 16.30 and time course data were plotted using GraphPad Prism 9.5.1.

### Cryo-EM data sample preparation and data collection

For the ACLY-D1026A−substrates sample, the protein was incubated with 5 mM CoA, 5 mM citrate, and 5 mM ATP at room temperature for 15 min.; and for the ACLY-D1026A−products sample, the protein was incubated with 5 mM acetyl-CoA, 5 mM OAA, and 5 mM ATP at room temperature for 15 min. About 1 μL 0.1%(m/v) $n$-dodecyl β-D-maltoside or 1 μL 0.05%(v/v) NP-40 was added to 20 μL protein sample and 3 μL of the protein/detergent mixture was applied to glow discharged Quantifoil R1.2/1.3 300 mesh copper grids with carbon foil and blotted for 10 s (blot force = 2) under 100% humidity at 16 °C and plunged into liquid ethane using a FEVitribot Mark IV. These grids were screened using an FEI TF20 microscope equipped with Falcon 3 detector. Data collection was performed using a Titan Krios equipped with a K3 direct detector (Gatan). Defocus range of 1.0–2.0 μm was applied in image acquisition with EPU 2.10.

### Image processing

All cryo-EM data were processed using the following workflow. First, beam-induced motions were corrected using MotionCor2[31]. After motion correction, all micrographs were imported into Cryosparc2 and subsequent data processing was performed in Cryosparc2[32]. Ctffind4 was used to determine the defocus value of each micrograph[33]. Micrographs, which contained CTF information worse than 5 Å, were discarded.

For 3D reconstruction of the ACLY-D1026A−substrates structure, 2,994,064 particles were picked from 5674 micrographs. These particles were classified into 200 classes. After 2D classification, 1,874,207 particles were kept. Four initial models were generated from these particles, and multiple rounds of heterogeneous refinement resulted in one symmetric tetramer with D2 symmetry and three asymmetric tetramers with distinct conformations. There were 378,979 particles in the class of the symmetric tetramer. A map with an overall resolution of 2.4 Å was generated from these particles after applying D2 symmetry. The resolution was further improved to 2.2 Å after global CTF refinement, local CTF refinement, and non-uniform refinement. Due to its flexibility, the local resolution of the ASH domain is lower than the CSH domain. Particle symmetry expansion with D2 symmetry was performed by rotating all four ASH monomers to put them in the same position. Particle subtraction was done to remove the duplicates of ASH monomers. By applying a mask covering the CSH tetramer and ASH monomer, 3D classification without alignment was performed. Particles showing poor density of (3 S)-citryl-CoA were discarded. About 1,122,938 particles were kept, and a 2.2 Å resolution map with an improved density of the ASH domain was obtained after local refinement. For the asymmetric particles, two maps with different conformations (ACLY-D1026A−substrates-asym-int and ACLY-D1026A−substrates-asym in Table 1) were generated from these particles with resolutions of 2.6 and 2.7 Å, respectively. Overlay of the three maps revealed that the asymmetric ASH domain rotates by about 40 degrees to increase the exposure to the solvent of the adjacent CSH domain. The asym-int structure shows rotations of the ASH domain of about 8 degrees. We, therefore, assigned the asym-int structure as intermediate conformations between the symmetric class and asymmetric class (Supplementary Fig. S2B).

For 3D reconstruction of the ACLY-D1026A−products structure, 2,126,238 particles were picked from 6057 micrographs. After 2D classification, 939,097 particles were kept. Four initial models were generated from these particles, and multiple rounds of heterogeneous refinement resulted in one symmetric tetramer with D2 symmetry, one asymmetric tetramer, and bad particles. There were 289,796 particles in the class of the symmetric tetramer and 183,036 in the asymmetric class. A map with an overall resolution of 2.7 Å was generated from these symmetric particles after applying D2 symmetry. The resolution was further improved to 2.4 Å after global CTF refinement, local CTF refinement, and non-uniform refinement. Due to its flexibility, the local resolution of the ASH domain is lower than the CSH domain. Particle symmetry expansion, particle subtraction, and 3D classification without alignment was performed to further sort these particles and 613,268 particles that showed a better density of (3 S)-citryl-CoA were

kept. A 2.5 Å resolution map with an improved density of the ASH domain was obtained after local refinement. For the asymmetric particles, a map at a resolution of 2.8 Å was generated. Image processing statistics for all cryo-EM data are provided in Table 1. For model building, the crystal structure of ACLY (PDB code: 6uuw)[24] was fitted into the cryo-EM maps by rigid body fitting. The atomic coordinates were manually adjusted in Coot 8.9.1[34] and refined against these maps by real-space-refinement in PHENIX 1.9[35]. All representations of cryo-EM density and structural models were prepared with Chimera 1.12[36] and Pymol (v1.20), DeLano Scientific, 2002, respectively.

### Reporting summary
Further information on research design is available in the Nature Portfolio Reporting Summary linked to this article.

## Data availability
Structures and EM maps of structures ACLY-D1026A−substrates (PDB 7RIG, EMD-24479), ACLY-D1026A−substrates, local refinement of ASH domain (PDB 8G5C, EMD-29739), ACLY-D1026A−substrates-asym-int (PDB 7RKZ, EMD-24511), ACLY-D1026A−substrates-asym (PDB 7RMP, EMD-24577), ACLY-D1026A−products (PDB 8G1F, EMD-29669), ACLY-D1026A−products, local refinement of ASH domain (PDB 8G5D, EMD-29670), and ACLY-D1026A−products-asym (PDB 8G1E, EMD-29668) have been deposited to the PDB and EMDB. Previously published structures and EM maps used in this study are available under the following accession codes: ACLY-WT with citrate and CoA (PDB 6UUZ, EMD-20903), ACLY-WT with acetyl-CoA and oxaloacetate and refined with D2 symmetry (PDB 6U19, EMD-20783), ACLY-WT with acetyl-CoA and oxaloacetate and refined with C1 symmetry (PDB 6UV5, EMD-20904), and ACLY-E599Q with ATP, citrate and CoA refined with D2 symmetry (PDB 6UV5, EMD-20902). Source data are provided with this paper.

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

## Acknowledgements

We thank B. Zuo and S. Molugu at the University of Pennsylvania Electron Microscopy Resource Laboratory for their help with negative staining and cryo-EM grids screening. We thank S. Steimle and D. Johnson-McDaniel at the University of Pennsylvania Beckman Center for Cryo-Electron Microscopy for their help in cryo-EM data collection, where most of the cryo-EM data collection was carried out. We thank N. Meyer and other staff members from the Pacific Northeast cryo-EM Center (PNCC) for their help in cryo-EM data collection. We thank T. Edwards, A. Wier, H. Wang, and other staff members from the National Cryo-EM facility (NCEF) for their help in cryo-EM data collection. Molecular graphics and structural analyses were performed with UCSF Chimera, developed by the Resource for Biocomputing, Visualization, and Informatics at the University of California, San Francisco, with support from NIH P41-GM103311. This work was supported by NIH grants R35 GM118090 and P01 AG031862 to R.M. and NIH grant R01 GM132261 to N.W.S.

## Author contributions

Conceptualization, X.W., H.L.P., E.M., A.V., N.W.S., and R.M.; Methodology, X.W., K.S., H.L.P., E.M., A.V., N.W.S., and R.M.; Investigation, X.W., K.S., H.L.P., E.M., and A.V.; Formal analysis, X.W., K.S., H.L.P., E.M., and A.V; Writing—original draft and visualization, X.W.; Writing—review and editing, X.W., K.S., H.L.P., E.M., A.V., N.W.S., and R.M.; Funding acquisition; R.M. and N.W.S.; Resources, R.M. and N.W.S.; Supervision, R.M. and N.W.S.

## Competing interests

The authors declare no competing interests.
