## [Peer Review File · Nature Communications]

REVIEWER COMMENTS

Reviewer #1 (Remarks to the Author):

This manuscript describes a series of structures of D1026A variant of ATP-citrate lyase (ACLY) and is a follow up work to the article that was published last year by the same group. The manuscript aims to dissect the role of ASH and CSH domains in catalysis. There has been much debate about whether the retro-aldol cleavage of citryl-CoA to acetyl-CoA takes place at the ASH or at the CSH domain.

It is remarkable that the authors have solved several cryoEM structures at high resolution and visualize the substrates and products in these structures. The data does show that there may be allosteric regulation. The main concern is that this still does not completely rule out the possibility of CSH domain taking part in the catalysis in the wild type ACLY.

The mutation of D1026 to Ala resulted in an additional non-canonical CoA binding site. The pantetheine arm of this 'non-canonical' binding site matches with that of the 'non-productive' CoA binding site observed in the wild type ACLY. It is not clear what makes the D1026A variant accommodate the CoA in the non-canonical binding site. Are there any structural changes (apart from the R976 side chain conformation change) associated with the CSH and/or ASH domain? Why is this binding mode not possible in wild type ACLY?

The experiments with saturated levels of acetyl-CoA and OAA does show that the reverse reaction takes place at ASH. The saturating conditions with D1026A variant may not resemble the WT reaction conditions. Further, this still does not rule out if the reaction can take place at the CSH domain or not. If all the reaction takes place at ASH domain, are the residues involved in the formation of citryl-CoA and conversion of citryl-CoA to acetyl-CoA same in the ASH domain? Have they been identified? In the 'other model' D1026 is proposed to be the key residue at the CSH domain to catalyze the conversion of citryl-CoA to acetyl-CoA. To clearly rule out that the CSH domain is not involved in catalysis, it is critical to identify the residues involved in the catalysis of Citryl-CoA to acetyl-CoA conversion in the ASH domain.

The conformation of F347 being different in citryl-CoA bound structure of D1026A variant and acetyl-CoA bound structure of WT is proposed as the reason for the conversion not taking place in the ASH domain. The conformation of pantetheine arm is also slightly different (Fig 3B). The conformational changes associated with F347 are indeed reversible based on Figure 4D.

In the binding studies with CoA, Fig 6A, the initial part of the binding curve looks almost sigmoidal like in Fig 6B. Does it somehow correlate with the proposal that the CoA initially binds at the 'non-productive' mode and then moves to ASH domain? Are these considered as two binding events or a single event?

In the mass spec analysis of wild type enzyme, no citryl-CoA was observed. Ideally, shouldn't there be some population of enzyme with citryl-CoA at any given point?

Too many different terms used for the CoA binding, Non-canonical, non-productive, loaded, unloaded, ejected etc. At some point, it becomes too difficult to understand the difference between these terms. The authors could use a much simpler terminology to define the CoA binding sites. Also, a figure giving an overview of all these binding sites would help understand the reader better.

Minor comments

Line 115: 'non-canonical CoA molecule is...' should be 'non-canonical CoA molecule in...'

The μL units has not been used uniformly in the text. μl , uL , ul etc.

Fig. 3F, CoA could be in the same orientation in Fig. 3D

Reviewer #2 (Remarks to the Author):

The manuscript by Wei et al. is structural and biochemical study of the catalytic mechanism underlying ATP citrate lyase (ACLY), an enzyme vital for the generation of acetyl-CoA. ACLY is composed of two domains: (1) the citrate synthase homology (CSH) domain, which promotes tetramerization; and (2) the acyl-CoA synthetase homology (ASH) domain, which binds ATP & citrate. CoA binding occurs at the interface of these two domains, of which two main sites have been previously described: (1) a 'productive' site, where the pantetheine arm extends into the ASH domain; and (2) a 'non-productive' site, where the arm is bent backwards towards the CSH domain. Mutations at both sites have proven to be vital for enzyme function, leaving open questions as to

catalytic mechanism – for example, if formation of the (3S)-citryl-CoA intermediate occurs at one site, while breakdown into acetyl-CoA & oxaloacetate (OAA) occurs at the other; or if one site serves as the loading/unloading platform and the other for catalysis.

To elucidate the roles of the CSH and ASH domains in the catalytic cycle, Wei et al. introduced an inactivating mutation on the CSH domain (D1026A) and performed a series of structural studies with both substrates and products. As this mutation targets the ‘non-productive’ site, they anticipated that the structural outputs they observe would shed light as to the role of this site, and the CSH domain itself, on catalytic function. Key results in this paper are: (i) determination of ACLYD1026A-substrate structures pre-incubated at cold and warm temperatures, which revealed (particularly for the warm temperature) density for both ‘non-productive’ CoA binding and formation of a (3S)-citryl-CoA intermediate in the ‘productive’ CoA binding site; (ii) determination of an ACLYD1026A-product structure, which revealed density for both acetyl-CoA & OAA at the ‘non-productive’ site and formation of a (3S)-citryl-CoA intermediate in the ‘productive’ CoA binding site; (iii) confirmation by mass spectrometry that the ACLYD1026A-product mixture can form (3S)-citryl-CoA intermediates, supporting their assignment by the EM density and biochemically demonstrating the mutant enzyme is not devoid of activity and can, in fact, form the catalytic intermediate from product molecules; and (iv) ITC binding data showing that ACLYD1026A, but not wild-type, appears to have a two-site CoA binding mode, with nearly 10 times higher affinity.

In general, the data are of high quality and the results are sound. Their biochemical follow up clearly answers the principal question put forward by the paper: what is the role of CSH in ACLY catalysis? They show that mutation of the CSH residue D1026A is not necessary for direct catalysis (as the mutant enzyme can form the catalytic intermediate from both substrate and products) and must instead play an allosteric or substrate-loading/unloading role.

Although mostly well presented, I feel that the manuscript could be improved with a few minor adjustments to help with clarity and some minor supporting experiments to delve a little further as to the mechanism by which this mutant is inactivating product formation. Addressing all or most of these comments would make it suitable for publication:

Major comments:

- The ‘cold’ dataset seems to serve little purpose in addressing the main points of the manuscript and is mostly confusing for the reader to navigate its value. As the ‘products’ and ‘warm substrates’ datasets are enough to make all points driven home by this paper, I suggest clarifying the story by focusing on those two and either pushing the ‘cold’ dataset to the supplementary or leave it out entirely.

- If the 'cold' dataset is to remain in the manuscript: in Figure 2C it looks like only a small part of CoA density is present. Furthermore, R976 is physically blocking nonproductive CoA binding as pointed out by the authors. A simple explanation for this seeming discrepancy is that a mixture of bound and unbound particles are being superimposed, leading to a hybrid reconstruction. Try seeing if symmetry-expanded particles can be sorted out into discrete bound or free states by focused classification with signal subtraction around the 'non-productive' CoA binding site. Only if efforts towards this fail, should the authors suggest CoA is only loosely/partially bound.

- It is not explicitly stated in the discussion as to how altered binding affinity generated by the D1026A mutation is contributing to inhibitory activity. Please expand on this a bit further: do they think that there is only one path into the ASH active site and that the increased binding affinity effectively 'gums up' the enzyme? In all cases where an (3S)-citryl-CoA intermediate is observed, there is a bound CoA factor at the 'non-productive' site. Does binding at the non-productive site stimulate formation of the catalytic intermediate, thus higher affinity prevents resolution of the catalytic cycle?

- Although they show in the wild-type context the likely main binding site is in the ASH domain (via a R576A mutant), it is not clear that it is also true for the D1026A mutant, which seems to have a large impact on CoA binding – most likely at the 'non-productive' site. Coupling the R576A mutation with D1026A would help to identify if the mutant is increasing affinity of the 'non-productive' or 'productive' site as part of its mechanism of inhibition.

- Can the authors propose mutations that specifically disrupt non-productive CoA binding to CSH domain while preserving binding of CoA to the ASH domain based on the structural models? An engineered ACYL where CoA can only bind to ASH domain can be incubated with products (i.e. CoA, citrate, Mg-ATP) and analyzed via LC-MS. If Model 2 (Fig 1) is correct, then the engineered ACYL should not produce any products or intermediates. However, if model 1 is correct, the engineered ACYL should produce (3S)-citryl-CoA intermediate, which may be detectable via mass spectrometry. The authors may consider discussing this as a future experiment in the discussion section of the manuscript.

Minor comments:

- Consider showing the rough location of Asp1026 in the CSH domain in Figure 1 to help the reader quickly gauge its position in the catalytic steps (similar to parts of fig 7, for example)

- A figure clearly demonstrating the two CoA binding modes superimposed with the relevant key residues shown (such as Wei ... Marmorstein 2020, Fig. 1E; but with side chains) would help the reader more quickly orient to the key question at play

- Line 96: please indicate the RMSD between both structures as no superposition of both reconstructions/models are actually shown in any of the figures

- Typo @ line 115: "...non-canonical CoA molecule is the ACLY-D1026A ..." -> 'is' should be replaced by 'in'

- Typo @ line 125: “therefor” should be replaced by “therefore”
- Wrong PDB reference in text? @ line 141: (7LI9) -> should be (6UV5)?
- What grids were used in this study? On line #520 they indicate Quantifoil R1.2/1.3 300 mesh copper grids – but what foil? Probably carbon – if so, please add this detail.
- Methods: how many movies were taken for each dataset? Only indicated is number of particles on the table 1, and only one dataset is given the movie number in the methods.
- Please include mask-corrected Fourier shell correlation (FSC) curves for all refined 3D cryoEM density maps in the supplementary data. Please also included particle image orientation distribution for the refined maps as well.
- Please report EMRinger scores (PMID: 26280328. Nat Methods. 2015 Oct; 12(10): 943–946) for the fitted models in table 1.
- Please report what species is the ACYL gene from, in the first paragraph of the method section.
- Typo @ line 488, “Naoh” should be “NaOH”.
- The authors may consider local followed by global CTF refinements in cryoSPARC to improve the resolution and quality of the maps even further.

Reviewer #3 (Remarks to the Author):

The manuscript by Wei et al describes allostery elicited by non-canonical binding of the substrate CoA in a domain of a mutant of ATP-citrate lyase (ACLY), which may catalyse formation of the reaction intermediate citryl-CoA. The authors utilize cryo-EM for structural characterization, existing X-ray crystal structures, and ligand binding measured by ITC, to ascertain conformational changes induced by CoA binding within a single monomer and within the homotetramer, along the catalytic pathway. Findings in this work, as well as previous work, demonstrate that a mutation of Asp1026 in the citrate synthase homology module (CSH) to an alanine leads to locking the catalytic pathway at stage in which the reaction intermediate (3S)-citryl-CoA does not progress to oxaloacetate and CoASH products. Two models are proposed for the apparent allosteric role of the substrate CoA.

This reviewer would be pleased to re-review this manuscript after significant revision. Major concerns are as follows:

(1) Data in Figure 2. Central to the research in this paper is the formation of citryl-CoA in the active site of the D1026A mutant of ACLY. It makes perfect sense that after addition of MgATP, citrate, and CoA to the enzyme solution that citryl-phosphate would form, CoA would displace it, but the absence of the apparent general base of Asp-1026 would prevent the retro-aldol reaction to produce oxaloacetate, phosphate and Ac-CoA as products. This requires divalent magnesium ion,

which is not mentioned to be in the reaction mixture. In Figures 2C, D, and 2F, the density between the thio-group of CoA and citrate is poorly defined, and does not suggest a thio-ester bond. How clear is it that this is not simply citrate and CoA binding proximal to each other, rather than being the citryl-CoA adduct? One expects that the citryl-CoA intermediate remains in the D1026A mutant form of ACLY because the retro-aldol reaction cannot occur. Why is the position of Asp-1026 or Ala-1026 not shown in any structure to give a sense of its proximity to the thio-ester of citryl-CoA?

(2) Line 176: The ACLY-D1026A mutant was incubated with the reaction products Ac-CoA and oxaloacetate to attempt to generate structures of the catalytic intermediates from the back reaction. But the authors cite “ATP” added to this and not ADP. Is this an error in the text? If not, how does this lead to the reverse reaction? Was phosphate present?

(3) Lines 214-229: Isothermal calorimetry reports on ligand or substrate binding to a macromolecule, but provides no details as to whether or not these binding events are catalytically competent. Does CoA bind to the WT and D1026A mutant ACLYs without other substrates or products binding first? Why did the investigators not conduct kinetic analysis of the wild-type and mutant enzymes?

(4) Lines 278- The cited non-productive binding of oxaloacetate observed in the cryo-EM studies should be observable as substrate inhibition of kinetics of the reverse reaction. Was that observed?

(5) Lines 271-274 “Interestingly, our metabolomic studies of ACLY-D1026A with acetyl-CoA and OAA products does not form (3S)-citryl-CoA in the absence of ATP. Although the reason for this is unclear, it is possible that ATP contributes to properly configuring the ASH domain for catalysis.” Was phosphate present, and would not it be needed to initiate catalysis of the reverse reaction? What role would ATP play in the reverse reaction?

(6) Lines 485-492. Studies of the catalytic mechanism of mammalian ACLYs indicate that MgATP is the first substrate to bind to enzyme, it then transfer its gamma-phosphate to His-760, ADP leaves, and only then may citrate or CoA bind (see Plowman and Cleland, 1967; Walsh and Spector, 1969). For the ITC samples involving both wild-type ACLY and its mutants, the enzymes are in a reaction mixture containing both 10 mM concentrations of phosphate, citrate, and magnesium chloride, with then added amounts of CoA. There is neither ATP nor ADP present. Does citrate or CoA bind to ACLY (wild-type or not) unless His-760 has first been N-phosphorylated? If these ligands do bind without this step, does the data reveal anything about the chemical mechanism of citryl-CoA formation?

(7) Lines 513-518: The Cryo-EM data for the ACLY-D1026A mutant was obtained in a solution of 5mM each of ATP, citrate, and CoA, but there is no mention of the presence of magnesium ion here.

While 5 mM MgCl₂ is mentioned as being in the enzyme storage buffer, what was its concentration on this study since no citryl-CoA can form with it being present?

Response to Reviewer Comments

Reviewer #1 (Remarks to the Author):

This manuscript describes a series of structures of D1026A variant of ATP-citrate lyase (ACLY) and is a follow up work to the article that was published last year by the same group. The manuscript aims to dissect the role of ASH and CSH domains in catalysis. There has been much debate about whether the retro-aldol cleavage of citryl-CoA to acetyl-CoA takes place at the ASH or at the CSH domain.

1.1. It is remarkable that the authors have solved several cryoEM structures at high resolution and visualize the substrates and products in these structures. The data does show that there may be allosteric regulation. The main concern is that this still does not completely rule out the possibility of CSH domain taking part in the catalysis in the wild type ACLY.

We agree that this study does leave open the possibility that the CSH domain could also directly participate in catalysis in wild type ACLY, although we believe that our study demonstrates that it does not play an essential catalytic role as the ASH is able to carry out the complete catalytic cycle. We now discuss this possibility in the discussion section of the revised manuscript.

We also provide additional data, described below, where we demonstrate that another mutant in the CSH domain that is predicted to disrupt non-productive CoA binding, R976E, yet still retains D1026, also produces detectable citryl-CoA from reaction products. We believe that this further demonstrates the non-essential catalytic role of residue D1026.

1.2. The mutation of D1026 to Ala resulted in an additional non-canonical CoA binding site. The pantetheine arm of this 'non-canonical' binding site matches with that of the 'non-productive' CoA binding site observed in the wild type ACLY. It is not clear what makes the D1026A variant accommodate the CoA in the non-canonical binding site. Are there any structural changes (apart from the R976 side chain conformation change) associated with the CSH and/or ASH domain? Why is this binding mode not possible in wild type ACLY?

For the binding of wild-type ACLY to non-productive CoA, residue D1026 makes an important water-mediated interaction with the sulfur atom of CoA that also involves residues H900 and R1065. We hypothesize that disruption of this network of water mediated interactions through the D1026A mutation, destabilizes non-productive CoA binding, thus making the pocket formed by residues H900, D1026 and R1065 available for mutually exclusive binding of non-canonical CoA. This is now described in the Results section (ACLY-D1026A with substrates traps (3S)-citryl-CoA in the ASH domain) of the revised manuscript.

1.3. The experiments with saturated levels of acetyl-CoA and OAA does show that the reverse reaction takes place at ASH. The saturating conditions with D1026A variant may not resemble the WT reaction conditions. Further, this still does not rule out if the reaction can take place at the CSH domain or not. If all the reaction takes place at ASH domain, are the residues involved in the formation of citryl-CoA and conversion of citryl-CoA to acetyl-CoA same in the ASH domain? Have they been identified? In the 'other model' D1026 is proposed to be the key residue at the CSH domain to catalyze the conversion of citryl-CoA to acetyl-CoA. To clearly rule out that the CSH domain is not involved in catalysis, it is critical to identify the residues involved in the catalysis of Citryl-CoA to acetyl-CoA conversion in the ASH

domain.

In Wei et al. (*Nature Struct. Mol. Biol.* 27: 33-, 2020) we demonstrated that residue E599 in the ASH domain plays an important catalytic role for generating acetyl-CoA and OAA products, as this mutant showed background levels of activity when mutated to alanine. We also propose that E599 might play a role as a general base for cleavage of citryl-CoA to acetyl-CoA, as there is no other residue in proximity to carry out this function. We now discuss this in the discussion section of the revised manuscript.

To further demonstrate that the CSH domain plays an allosteric role instead of a direct catalytic role in cleavage of citryl-CoA to acetyl CoA, we evaluated the ability of another ACLY mutant that is predicted to disrupt non-productive CoA binding within the CSH domain, ACLY-R976E (Wei et al. *Nature Struct. Mol. Biol.* 27: 33-, 2020), for its ability to convert products to citryl-CoA. Like ACLY-D1026A, ACLY-R976E was able to generate citryl-CoA (even in the presence of residue D1026). Together, we now demonstrate that two different mutants that are predicted to disrupt non-productive CoA binding produce detectable citryl-CoA intermediates from reaction products. We believe that this data argues even more strongly that D1026 does not play an essential catalytic role in cleavage of citryl-CoA to acetyl-CoA. This new data is now shown in Supplemental Figure 6 and described in the results and discussion sections of the revised manuscript.

1.4. The conformation of F347 being different in citryl-CoA bound structure of D1026A variant and acetyl-CoA bound structure of WT is proposed as the reason for the conversion not taking place in the ASH domain. The conformation of pantetheine arm is also slightly different (Fig 3B). The conformational changes associated with F347 are indeed reversible based on Figure 4D.

The conformational change of F347 is indeed reversible, and this is now described in the discussion section of the revised manuscript.

1.5. In the binding studies with CoA, Fig 6A, the initial part of the binding curve looks almost sigmoidal like in Fig 6B. Does it somehow correlate with the proposal that the CoA initially binds at the 'non-productive' mode and then moves to ASH domain? Are these considered as two binding events or a single event?

In order to address this concern, we repeated our ITC experiments and observed that during the ITC experiments, both the ACLY-WT and ACLY-D1026A mutant precipitated during the experiment regardless of whether buffer only or CoA was titrated into the reaction chamber. We repeated the ITC experiments several times under different conditions, including conditions that were previously published (Sun et al. *J. Biol. Chem.* 285: 27418-, 2010) but could not avoid the protein precipitation that we observed. We suspect that protein precipitation also occurred in our initial titrations, but we did not notice it. Because of this protein precipitation, we do not believe that we can make reliable conclusions about CoA binding properties to ACLY and mutants and have therefore removed the ITC data (former Figure 6). We do not believe that exclusion of the ITC data changes the main conclusions of the current study.

1.6. In the mass spec analysis of wild type enzyme, no citryl-CoA was observed. Ideally, shouldn't there be some population of enzyme with citryl-CoA at any given point?

We suspect that the reaction in the wild-type enzyme with products goes to completion to CoA. Of note, we do not routinely detect citryl-CoA from cell extracts and to our knowledge existing literature does not describe detection of a stable pool of cellular citryl-CoA in metazoans. Thus, we agree with the reviewer's suspicion that there may be some citryl-CoA produced by/bound to the wild-type enzyme, however it is not detectable in our hands. We have added a comment regarding our belief that the ACLY-WT does not form a stable citryl-CoA intermediate in the results (ACLY-D1026A with products is able to form (3S)-citryl-CoA in the ASH domain section of the revised manuscript.

1.7. Too many different terms used for the CoA binding, Non-canonical, non-productive, loaded, unloaded, ejected etc. At some point, it becomes too difficult to understand the difference between these terms. The authors could use a much simpler terminology to define the CoA binding sites. Also, a figure giving an overview of all these binding sites would help understand the reader better.

In the revised manuscript, we now only use non-canonical, non-productive (loading), productive and unloading. For additional clarity, we also indicate these states in revised Figures 1 and 7A.

1.8. Minor comments

Line 115: 'non-canonical CoA molecule is...' should be 'non-canonical CoA molecule in...'

This has been corrected in the revised manuscript.

The μL units has not been used uniformly in the text. μl , uL , ul etc.

This has been corrected in the revised manuscript.

Fig. 3F, CoA could be in the same orientation in Fig. 3D

We assume that the reviewer is refereeing to Figure 2 since there is not panel 3F. Panel 2E (formally 2F) has been rotated to correspond to the orientation of non-canonical CoA shown in current panel 2D.

Reviewer #2 (Remarks to the Author):

The manuscript by Wei et al. is structural and biochemical study of the catalytic mechanism underlying ATP citrate lyase (ACLY), an enzyme vital for the generation of acetyl-CoA. ACLY is composed of two domains: (1) the citrate synthase homology (CSH) domain, which promotes tetramerization; and (2) the acyl-CoA synthetase homology (ASH) domain, which binds ATP & citrate. CoA binding occurs at the interface of these two domains, of which two main sites have been previously described: (1) a 'productive' site, where the pantetheine arm extends into the ASH domain; and (2) a 'non-productive' site, where the arm is bent backwards towards the CSH domain. Mutations at both sites have proven to be vital for enzyme function, leaving open questions as to catalytic mechanism – for example, if formation of the (3S)-citryl-CoA intermediate occurs at one site, while breakdown into acetyl-CoA & oxaloacetate (OAA) occurs at the other; or if one site serves as the loading/unloading platform and the other for catalysis.

To elucidate the roles of the CSH and ASH domains in the catalytic cycle, Wei et al. introduced an inactivating mutation on the CSH domain (D1026A) and performed a series of structural studies with

both substrates and products. As this mutation targets the 'non-productive' site, they anticipated that the structural outputs they observe would shed light as to the role of this site, and the CSH domain itself, on catalytic function. Key results in this paper are: (i) determination of ACLYD1026A-substrate structures pre-incubated at cold and warm temperatures, which revealed (particularly for the warm temperature) density for both 'non-productive' CoA binding and formation of a (3S)-citryl-CoA intermediate in the 'productive' CoA binding site; (ii) determination of an ACLYD1026A-product structure, which revealed density for both acetyl-CoA & OAA at the 'non-productive' site and formation of a (3S)-citryl-CoA intermediate in the 'productive' CoA binding site; (iii) confirmation by mass spectrometry that the ACLYD1026A-product mixture can form (3S)-citryl-CoA intermediates, supporting their assignment by the EM density and biochemically demonstrating the mutant enzyme is not devoid of activity and can, in fact, form the catalytic intermediate from product molecules; and (iv) ITC binding data showing that ACLYD1026A, but not wild-type, appears to have a two-site CoA binding mode, with nearly 10 times higher affinity.

In general, the data are of high quality and the results are sound. Their biochemical follow up clearly answers the principal question put forward by the paper: what is the role of CSH in ACLY catalysis? They show that mutation of the CSH residue D1026A is not necessary for direct catalysis (as the mutant enzyme can form the catalytic intermediate from both substrate and products) and must instead play an allosteric or substrate-loading/unloading role.

Although mostly well presented, I feel that the manuscript could be improved with a few minor adjustments to help with clarity and some minor supporting experiments to delve a little further as to the mechanism by which this mutant is inactivating product formation. Addressing all or most of these comments would make it suitable for publication:

Major comments:

2.1. The 'cold' dataset seems to serve little purpose in addressing the main points of the manuscript and is mostly confusing for the reader to navigate its value. As the 'products' and 'warm substrates' datasets are enough to make all points driven home by this paper, I suggest clarifying the story by focusing on those two and either pushing the 'cold' dataset to the supplementary or leave it out entirely.

As suggested by the reviewer, for clarity of presentation, we have removed the 'cold' data from the revised manuscript. Comparisons to the cold structure have been replaced by comparisons to the ACLY-WT structure with CoA (PDB 6UUZ) and to the asymmetric ACLY-D1026A—substrate structure.

2.2. If the 'cold' dataset is to remain in the manuscript: in Figure 2C it looks like only a small part of CoA density is present. Furthermore, R976 is physically blocking nonproductive CoA binding as pointed out by the authors. A simple explanation for this seeming discrepancy is that a mixture of bound and unbound particles are being superimposed, leading to a hybrid reconstruction. Try seeing if symmetry-expanded particles can be sorted out into discrete bound or free states by focused classification with signal subtraction around the 'non-productive' CoA binding site. Only if efforts towards this fail, should the authors suggest CoA is only loosely/partially bound.

As indicated above, we have removed the 'cold' data from the revised manuscript.

2.3. It is not explicitly stated in the discussion as to how altered binding affinity generated by the

D1026A mutation is contributing to inhibitory activity. Please expand on this a bit further: do they think that there is only one path into the ASH active site and that the increased binding affinity effectively 'gums up' the enzyme? In all cases where an (3S)-citryl-CoA intermediate is observed, there is a bound CoA factor at the 'non-productive' site. Does binding at the non-productive site stimulate formation of the catalytic intermediate, thus higher affinity prevents resolution of the catalytic cycle?

As discussed in our response to reviewer 1 (response to 1.5), in our repeat ITC titrations we observed that both ACLY-WT and the ACLY-D1026A mutant precipitated during the experiment regardless of whether buffer only or CoA was titrated into the reaction chamber, and precipitate could not be avoided when we altered buffer or salt conditions or employed previously published conditions (Sun et al. *J. Biol. Chem.* 285: 27418-, 2010). Because of this, we do not believe that we can make reliable conclusions about CoA binding properties to ACLY and mutants and have therefore removed all of the ITC data from the revised manuscript. We do not believe that exclusion of the ITC data changes the main conclusions of the current study.

Our data demonstrates that the D1026A mutation in the CSH domain results in a CoA-bound ASH domain that is not competent for transformation to acetyl-CoA in the ASH domain. We propose that this occurs because of improper CoA loading from the non-productive CSH-bound CoA to productive ASH-bound CoA. This issue is now addressed in the Discussion sections of the revised manuscript.

2.3. Although they show in the wild-type context the likely main binding site is in the ASH domain (via a R576A mutant), it is not clear that it is also true for the D1026A mutant, which seems to have a large impact on CoA binding – most likely at the 'non-productive' site. Coupling the R576A mutation with D1026A would help to identify if the mutant is increasing affinity of the 'non-productive' or 'productive' site as part of its mechanism of inhibition.

At the suggestion of the reviewer, we prepared the ACLY-R576A/D1026A double mutant and attempted to evaluate CoA binding using ITC. However, as observed with ACLY-WT and ACLY-D1026A, we observed protein precipitation in the reaction chamber (see responses 1.5 and 2.2 above). Because of this we could not obtain reliable data for this double mutant so have not included data on this mutant in this study. Nonetheless, we believe that the absence of this data does not detract from the main conclusions of the study.

2.4. Can the authors propose mutations that specifically disrupt non-productive CoA binding to CSH domain while preserving binding of CoA to the ASH domain based on the structural models? An engineered ACYL where CoA can only bind to ASH domain can be incubated with products (i.e. CoA, citrate, Mg-ATP) and analyzed via LC-MS. If Model 2 (Fig 1) is correct, then the engineered ACYL should not produce any products or intermediates. However, if model 1 is correct, the engineered ACYL should produce (3S)-citryl-CoA intermediate, which may be detectable via mass spectrometry. The authors may consider discussing this as a future experiment in the discussion section of the manuscript.

As suggested by the reviewer, we prepared an additional ACLY mutant that is predicted to destabilize non-productive CoA binding but not effect productive CoA binding and incubated this mutant with products and carried out LC-MS studies to probe for production of (3S)-citryl-CoA intermediate. The mutant that we prepared was ACLY-R976E, as R976 was observed to interact with non-productive CoA (but not productive CoA) in the ACLY-WT bound to citrate and CoA refined without

symmetry in the closed conformation, and an ACLY-R976E was shown to have background levels of activity (Wei et al. *Nature Struct. Mol. Biol.* 27: 33-, 2020). Like ACLY-D1026A, we found that ACLY-R976E was able to generate citryl-CoA (even in the presence of residue D1026). This result is also consistent with an allosteric role of the CSH domain (model 2) since this mutant also disrupts proper CoA loading from the non-productive to productive conformations. Together, we now demonstrate that two different mutants that are predicted to disrupt non-productive CoA binding produce detectable (3S)-citryl-CoA intermediate. We believe that this data argues even more strongly that D1026 does not play an essential catalytic role in cleavage of citryl-CoA to acetyl-CoA. This data is now shown in Supplemental Figure 6 and described in the results and discussion sections of the revised manuscript.

2.5. Minor comments:

- Consider showing the rough location of Asp1026 in the CSH domain in Figure 1 to help the reader quickly gauge its position in the catalytic steps (similar to parts of fig 7, for example)

We have included the rough position of the D1026A mutation in revised panel 7A. We have also added another **figure panel 1B to Figure 1** comparing the productive and non-productive CoA binding modes highlighting key residues highlighted in this study D1026 and R976.

- A figure clearly demonstrating the two CoA binding modes superimposed with the relevant key residues shown (such as Wei ... Marmorstein 2020, Fig. 1E; but with side chains) would help the reader more quickly orient to the key question at play

Another figure panel 1B to Figure 1 has been added to compare the productive and non-productive CoA binding modes and highlighting key residues mutated in this study, D1026 and R976.

- Line 96: please indicate the RMSD between both structures as no superposition of both reconstructions/models are actually shown in any of the figures

At the request of reviewer 1, we have removed the 'cold' structure from the revised manuscript, so we now compare the ACLY-D1026A-substrates structure to the ACLY-WT structure with CoA (PDB 6UUZ) and an RMSD is provided for this comparison in the revised manuscript.

- Typo @ line 115: "...non-canonical CoA molecule is the ACLY-D1026A ..." -> 'is' should be replaced by 'in'

This has been corrected in the revised manuscript.

- Typo @ line 125: "therefor" should be replaced by "therefore"

This has been corrected in the revised manuscript.

- Wrong PDB reference in text? @ line 141: (7LI9) -> should be (6UV5)?

This has been corrected in the revised manuscript.

- What grids were used in this study? On line #520 they indicate Quantifoil R1.2/1.3 300 mesh copper grids – but what foil? Probably carbon – if so, please add this detail.

We now indicate that we use Quantifoil R1.2/1.3 300 mesh copper grids with carbon foil.

- Methods: how many were taken for each dataset? Only indicated is number of particles on the table 1, and only one dataset is given the movie number in the methods.

We now specify the number of movies used in Table 1.

- Please include mask-corrected Fourier shell correlation (FSC) curves for all refined 3D cryoEM density maps in the supplementary data. Please also included particle image orientation distribution for the refined maps as well.

This data is now included in Supplemental Data Figure 1B.

- Please report EMRinger scores (PMID: 26280328. Nat Methods. 2015 Oct; 12(10): 943–946) for the fitted models in table 1.

EMRinger scores are now included in Table1.

- Please report what species is the ACYL gene from, in the first paragraph of the method section.

We now indicate the ACLY gene used in this study is from human.

- Typo @ line 488, “Naoh” should be “NaOH”.

This has been corrected in the revised manuscript.

- The authors may consider local followed by global CTF refinements in cryoSPARC to improve the resolution and quality of the maps even further.

All EM maps reported in the manuscript were generated after global and local CTF refinement. This is indicated in the methods section of the revised manuscript.

Reviewer #3 (Remarks to the Author):

The manuscript by Wei et al describes allostery elicited by non-canonical binding of the substrate CoA in a domain of a mutant of ATP-citrate lyase (ACLY), which may catalyse formation of the reaction intermediate citryl-CoA. The authors utilize cryo-EM for structural characterization, existing X-ray crystal structures, and ligand binding measured by ITC, to ascertain conformational changes induced by CoA binding within a single monomer and within the homotetramer, along the catalytic pathway. Findings in this work, as well as previous work, demonstrate that a mutation of Asp1026 in the citrate synthase homology module (CSH) to an alanine leads to locking the catalytic pathway at stage in which the reaction intermediate (3S)-citryl-CoA does not progress to oxaloacetate and CoASH products. Two

models are proposed for the apparent allosteric role of the substrate CoA.

This reviewer would be pleased to re-review this manuscript after significant revision. Major concerns are as follows:

3.1. Data in Figure 2. Central to the research in this paper is the formation of citryl-CoA in the active site of the D1026A mutant of ACLY. It makes perfect sense that after addition of MgATP, citrate, and CoA to the enzyme solution that citryl-phosphate would form, CoA would displace it, but the absence of the apparent general base of Asp-1026 would prevent the retro-aldol reaction to produce oxaloacetate, phosphate and Ac-CoA as products. This requires divalent magnesium ion, which is not mentioned to be in the reaction mixture. In Figures 2C, D, and 2F, the density between the thio-group of CoA and citrate is poorly defined, and does not suggest a thio-ester bond. How clear is it that this is not simply citrate and CoA binding proximal to each other, rather than being the citryl-CoA adduct? One expects that the citryl-CoA intermediate remains in the D1026A mutant form of ACLY because the retro-aldol reaction cannot occur. Why is the position of Asp-1026 or Ala-1026 not shown in any structure to give a sense of its proximity to the thio-ester of citryl-CoA?

5 mM Mg⁺² was included in the protein preparation for structural analysis and LC-MS analysis of metabolites, and this information is now included in the results (ACLY-D1026A with products is able to form (3S)-citryl-CoA in the ASH domain) section of the revised manuscript.

We believe that the cryo-EM density shown in the ACLY-D1026A-substrates structure (Figure 2C) is most consistent with (3S)-citryl-CoA, rather than citrate + CoA, and we have now added **Extended Data Figure 2A** to further illustrate this point. In this figure panel, we compare the cryo-EM density of modeled (3S)-citryl-CoA from the ACLY-D1026A-substrates structure with the same density modeled as CoA + citrate.

3.2. Line 176: The ACLY-D1026A mutant was incubated with the reaction products Ac-CoA and oxaloacetate to attempt to generate structures of the catalytic intermediates from the back reaction. But the authors cite "ATP" added to this and not ADP. Is this an error in the text? If not, how does this lead to the reverse reaction? Was phosphate present?

For the preparation of ACLY-D1026A with products for cryo-EM studies and LC-MS analysis, we used ATP in place to ADP + phosphate to more closely mimic an active conformation of the H760-containing loop. Consistent with the requirement of this active conformation for chemistry, we now show new LC-MS data in **Extended Data Figure 5B**, illustrating that ACLY-D1026A with acetyl-CoA, OAA and either ATP or ATP-γS is able to form citryl-CoA, while ACLY-with ATP/ATPγS replaced with either ADP or ADP + phosphate cannot. We believe that this likely highlights the importance of a phosphate group and ordering of the H760 containing loop for catalysis, something that presumably cannot be suitably mimicked by ADP + phosphate. These points are now clarified in the results (ACLY-D1026A with products is able to form (3S)-citryl-CoA in the ASH domain) and discussion sections of the revised manuscript.

3.3. Lines 214-229: Isothermal calorimetry reports on ligand or substrate binding to a macromolecule, but provides no details as to whether or not these binding events are catalytically competent. Does CoA bind to the WT and D1026A mutant ACLYs without other substrates or products binding first? Why did the investigators not conduct kinetic analysis of the wild-type and mutant enzymes?

ACLY-D1026A does not show detectable catalytic activity so is not amenable to kinetic analysis.

3.4. Lines 278- The cited non-productive binding of oxaloacetate observed in the cryo-EM studies should be observable as substrate inhibition of kinetics of the reverse reaction. Was that observed?

We are currently carrying out studies to address the role of the non-productive OAA bound to the CSH domain and plan to present that analysis in a separate subsequent study.

3.5. Lines 271-274 “Interestingly, our metabolomic studies of ACLY-D1026A with acetyl-CoA and OAA products does not form (3S)-citryl-CoA in the absence of ATP. Although the reason for this is unclear, it is possible that ATP contributes to properly configuring the ASH domain for catalysis.” Was phosphate present, and would not it be needed to initiate catalysis of the reverse reaction? What role would ATP play in the reverse reaction?

As described in response to item 2 above, the phosphate group is provided by ATP in these studies in an attempt to better mimic the active conformation of the enzyme.

3.6. Lines 485-492. Studies of the catalytic mechanism of mammalian ACLYs indicate that MgATP is the first substrate to bind to enzyme, it then transfer its gamma-phosphate to His-760, ADP leaves, and only then may citrate or CoA bind (see Plowman and Cleland, 1967; Walsh and Spector, 1969). For the ITC samples involving both wild-type ACLY and its mutants, the enzymes are in a reaction mixture containing both 10 mM concentrations of phosphate, citrate, and magnesium chloride, with then added amounts of CoA. There is neither ATP nor ADP present. Does citrate or CoA bind to ACLY (wild-type or not) unless His-760 has first been N-phosphorylated? If these ligands do bind without this step, does the data reveal anything about the chemical mechanism of citryl-CoA formation?

Our ITC studies for analysis of CoA binding to ACLY-WT and mutants were initially carried out in the absence of ATP or ADP, to match prior studies demonstrating binding without ATP/ADP (Sun et al. *J. Biol. Chem.* 285: 27418-, 2010). However, on repeat of these experiments, we observed protein precipitation in the reaction chamber (see response 1.5 above). We also tried to carry out the ITC experiments in the presence of ATP or ADP but observed similar protein precipitation. Because of this protein precipitation, we do not believe that we can make reliable conclusions about CoA binding properties to ACLY and mutants and have therefore removed the ITC data (former Figure 6). We do not believe that exclusion of the ITC data changes the main conclusions of the current study.

3.7. Lines 513-518: The Cryo-EM data for the ACLY-D1026A mutant was obtained in a solution of 5mM each of ATP, citrate, and CoA, but there is no mention of the presence of magnesium ion here. While 5 mM MgCl₂ is mentioned as being in the enzyme storage buffer, what was its concentration on this study since no citryl-CoA can form with it being present?

5 mM Mg⁺² was included in the protein preparation for structural analysis and LC-MS analysis of metabolites, and this information is now included in the results (ACLY-D1026A with products is able to form (3S)-citryl-CoA in the ASH domain) section of the revised manuscript.

REVIEWER COMMENTS

Reviewer #1 (Remarks to the Author):

The revised manuscript 'Allosteric role of the citrate synthase homology domain of ATP citrate lyase' is now far more convincing and has more clarity. The authors have carefully considered the comments of reviewers and performed additional experiments to clarify the issues raised in the earlier version of the manuscript. I recommend the acceptance of this manuscript for publication.

Reviewer #2 (Remarks to the Author):

The revised manuscript by Wei et al. addresses our original major & minor comments (where possible). We support removal of the ITC binding data for the reasons given by the authors and believe that the additional analysis of an orthogonal mutant at the non-productive site (ACLY-R976E) serves to bridge some of the gap left behind by the loss of the binding data. Overall, we are of the opinion that the main conclusions of the revised manuscript are sound and support its publication with a few minor comments:

- (1.) The abstract text should be adjusted to remove reference to the removed binding data: <Line 35> "..., and exhibits altered cofactor binding...", and the sentence adjusted as necessary.
- (2.) Grammatical error at Line 116: "Notably, Arg 976, which plays an important role in binding non-productive, CoA changes..." -> probably meant to have comma after 'CoA'
- (3.) The density in supplemental Figure 2A could be shown at decreasing thresholds to better establish/show the continuous nature of the density for the intermediate.
- (4.) While we think it is not required for publication, the authors may want to explore alternative methods to measure binding that may be more amenable to their system if they are observing concentration-dependent aggregation/precipitation. Microscale thermophoresis may be a potential avenue that could potentially work at lower concentrations that may avoid precipitation problems.

Reviewer #3 (Remarks to the Author):

I have read the comments of all three reviewers and how the authors

have addressed these comments in their revised manuscript. Noteworthy among the responses are the repeated remarks that ITC data, which would describe the avidity of binding events for substrates to ACLY and its mutants was complicated by precipitation events, which is often observed at high concentrations of ligands in this mode of analysis, a fair point. What is lacking in this revision is the following: while there is demonstration that the critical intermediate citryl-CoA is fitted to the cryo-EM data, there is no evidence that this intermediate is, or is not, a kinetically competent intermediate in the reaction pathway. This reviewer has suggested that simple kinetic analysis be done on the wild-type and mutant forms of this protein to determine this, and this has not been addressed. So despite the high values of the structural data provided in this manuscript, without corresponding, and readily obtained, kinetic analysis, this reviewer does not think that this manuscript is as yet suitable for publication in Nature Communications

Response to Reviewer Comments

Reviewer #1 (Remarks to the Author):

The revised manuscript 'Allosteric role of the citrate synthase homology domain of ATP citrate lyase' is now far more convincing and has more clarity. The authors have carefully considered the comments of reviewers and performed additional experiments to clarify the issues raised in the earlier version of the manuscript. I recommend the acceptance of this manuscript for publication.

We are happy that we were able to satisfy the concerns of this reviewer.

Reviewer #2 (Remarks to the Author):

The revised manuscript by Wei et al. addresses our original major & minor comments (where possible). We support removal of the ITC binding data for the reasons given by the authors and believe that the additional analysis of an orthogonal mutant at the non-productive site (ACLY-R976E) serves to bridge some of the gap left behind by the loss of the binding data. Overall, we are of the opinion that the main conclusions of the revised manuscript are sound and support its publication with a few minor comments:

(1.) The abstract text should be adjusted to remove reference to the removed binding data: "..., and exhibits altered cofactor binding...", and the sentence adjusted as necessary.

We have removed the relevant text.

(2.) Grammatical error at Line 116: "Notably, Arg 976, which plays an important role in binding non-productive, CoA changes..." -> probably meant to have comma after 'CoA'

A comma has been added after CoA.

(3.) The density in supplemental Figure 2A could be shown at decreasing thresholds to better establish/show the continuous nature of the density for the intermediate.

In Figure 2A, we illustrate that the observed density is incompatible with CoA-citrate as the molecules are too close to not be covalently bonded. We hope that this satisfies the reviewer.

(4.) While we think it is not required for publication, the authors may want to explore alternative methods to measure binding that may be more amenable to their system if they are observing concentration-dependent aggregation/precipitation. Microscale thermophoresis may be a potential avenue that could potentially work at lower concentrations that may avoid precipitation problems.

We had tried alternative methods including fluorescence quenching but ran into other technical issues that could not be overcome. We had not explored microscale thermophoresis but decided not to since we felt that the binding data was no longer a significant part of the story.

Reviewer #3 (Remarks to the Author):

I have read the comments of all three reviewers and how the authors have addressed these comments in their revised manuscript. Noteworthy among the responses are the repeated remarks that ITC data, which would describe the avidity of binding events for substrates to ACLY and its mutants was complicated by precipitation events, which is often observed at high concentrations of ligands in this mode of analysis, a fair point. What is lacking in this revision is the following: while there is demonstration that the critical intermediate citryl-CoA is fitted to the cryo-EM data, there is no evidence that this intermediate is, or is not, a kinetically competent intermediate in the reaction pathway. This reviewer has suggested that simple kinetic analysis be done on the wild-type and mutant forms of this protein to determine this, and this has not been addressed. So despite the high values of the structural data provided in this manuscript, without corresponding, and readily obtained, kinetic analysis, this reviewer does not think that this manuscript is as yet suitable for publication in Nature Communications

The reviewer brings up a fair point. To demonstrate that citryl-CoA formation by the ACLY-D1026A mutant is a kinetically competent intermediate, we incubated ACLY-D1026A with acetyl-CoA and OAA products and ATP and 5 mM MgCl₂ at room temperature and monitored citryl-CoA formation over time by quenching the reaction and semi-quantitation of citryl-CoA formation by LC-HRMS and LC-MS/MS using ¹³C₃ ¹⁵N₁-HMG-CoA as a surrogate internal standard to derive relative abundance of citryl-CoA. As can be seen in our **Extended Data Figure 7A**, this analysis reveals a time-dependent increase of citryl-CoA formation with ACLY-D1026A but not with ACLY-WT. Analogous studies of ACLY-D1026A and ACLY-WT incubated with CoA, citrate and ATP substrates also reveals time dependent formation of citryl-CoA for ACLY-D1026A, but not for ACLY-WT (**Extended Data Figure 7B**). As expected, ACLY-WT, but not ACLY-D1026A, showed time dependent formation of acetyl-CoA. Although the error bars are admittedly large for some of the data points of this semi-quantitative analysis (likely due to an imperfect surrogate internal standard), we believe that these new studies now demonstrate that the ACLY-D1026A mutant produces a kinetically competent citryl-CoA intermediate when incubated with either CoA and citrate substrates or acetyl-CoA and OAA products.

We believe that this new data significantly strengthens the manuscript and addresses the concern of the review.

REVIEWERS' COMMENTS

Reviewer #3 (Remarks to the Author):

The manuscript by Wei et al. has been revised, and in its revisions have adequately addressed concerns raised by this reviewer in terms of the new application of kinetic analysis to elucidate the formation of a key reaction intermediate, citryl-CoASH in the mutant enzyme, as well as the establishment of its kinetic competence in the reaction pathway. Publication of this revised manuscript is now encouraged.